# Identification of leukemic and pre-leukemic stem cells by clonal tracking from single-cell transcriptomics

Lars Velten [1,2,17✉], Benjamin A. Story [3,4,17], Pablo Hernández-Malmierca [5,6,17], Simon Raffel [5,6,7], Daniel R. Leonce[3], Jennifer Milbank[3], Malte Paulsen[8], Aykut Demir [7], Chelsea Szu-Tu[1,2], Robert Frömel[1,2], Christoph Lutz[7], Daniel Nowak [9], Johann-Christoph Jann [9], Caroline Pabst[7,10], Tobias Boch[4,6,9], Wolf-Karsten Hofmann[9], Carsten Müller-Tidow [7], Andreas Trumpp [5,6,11], Simon Haas [5,6,11,12,13,14] & Lars M. Steinmetz[3,15,16✉]

Cancer stem cells drive disease progression and relapse in many types of cancer. Despite this, a thorough characterization of these cells remains elusive and with it the ability to eradicate cancer at its source. In acute myeloid leukemia (AML), leukemic stem cells (LSCs) underlie mortality but are difficult to isolate due to their low abundance and high similarity to healthy hematopoietic stem cells (HSCs). Here, we demonstrate that LSCs, HSCs, and pre-leukemic stem cells can be identified and molecularly profiled by combining single-cell transcriptomics with lineage tracing using both nuclear and mitochondrial somatic variants. While mutational status discriminates between healthy and cancerous cells, gene expression distinguishes stem cells and progenitor cell populations. Our approach enables the identification of LSC-specific gene expression programs and the characterization of differentiation blocks induced by leukemic mutations. Taken together, we demonstrate the power of single-cell multi-omic approaches in characterizing cancer stem cells.

---

[1] Centre for Genomic Regulation (CRG), The Barcelona Institute of Science and Technology, Barcelona, Spain. [2] Universitat Pompeu Fabra (UPF), Barcelona, Spain. [3] European Molecular Biology Laboratory (EMBL), Genome Biology Unit, Heidelberg, Germany. [4] Swiss Federal Institute of Technology (ETH) Zurich, Department of Biosystems Science and Engineering, Basel, Switzerland. [5] Heidelberg Institute for Stem Cell Technology and Experimental Medicine (HI-STEM gGmbH), Heidelberg, Germany. [6] Division of Stem Cells and Cancer, Deutsches Krebsforschungszentrum (DKFZ) and DKFZ-ZMBH Alliance, Heidelberg, Germany. [7] Department of Internal Medicine V, Hematology, Oncology and Rheumatology, University of Heidelberg, Heidelberg, Germany. [8] European Molecular Biology Laboratory (EMBL), Flow Cytometry Core Facility, Heidelberg, Germany. [9] Department of Hematology and Oncology, Medical Faculty Mannheim, Heidelberg University, Mannheim, Germany. [10] Molecular Medicine Partnership Unit (MMPU), University of Heidelberg and European Molecular Biology Laboratory (EMBL), Heidelberg, Germany. [11] German Cancer Consortium (DKTK), Heidelberg, Germany. [12] Berlin Institute of Health (BIH), Berlin, Germany. [13] Charité-Universitätsmedizin, Berlin, Germany. [14] Berlin Institute for Medical Systems Biology, Max Delbrück Center for Molecular Medicine in the Helmholtz Association, Berlin, Germany. [15] Department of Genetics, Stanford University School of Medicine, Stanford, CA, USA. [16] Stanford Genome Technology Center, Palo Alto, CA, USA. [17] These authors contributed equally: Lars Velten, Benjamin A. Story, Pablo Hernández-Malmierca. ✉email: lars.velten@crg.eu; larsms@embl.de

Tissues with high cellular turnover, such as the hematopoietic system or the intestine, depend on "professional" adult stem cells for their continuous regeneration[1]. Oncogenic mutations in these cells can cause cancers that maintain a hierarchical organization reminiscent of the tissue of origin. Only cancer stem cells (CSCs), residing at the top of the hierarchy, are able to fuel long-term cancer growth and drive relapse, whereas the bulk of the cancer consists of rapidly dividing cells with limited capacity for self-renewal, i.e., cells that exhaust their replicative potential after a finite number of divisions[2–4]. Owing to their stem cell-like properties, CSCs constitute an important driver of relapse, but their low division rates make them difficult to target therapeutically. Tools that permit the confident identification and characterization of CSCs are therefore urgently needed.

Acute myeloid leukemia (AML) serves as a paradigm for the study of cancer stem cells[5]. In 10–20% of healthy individuals over age 70, the acquisition of pre-leukemic mutations in hematopoietic stem cells (HSCs) results in the dominance of a small number of HSC-derived clones, a process termed Clonal Hematopoiesis of Indeterminate Potential (CHIP)[6,7]. While such pre-leukemic stem cells (pre-LSCs) are capable of giving rise to healthy blood and immune cells, additional mutations can cause a complete block in differentiation and thereby result in the malignant expansion of aberrant progenitor cells[8]. The accumulation of these so-called "blast" cells is ultimately fueled by the presence of leukemic stem cells (LSCs). Classic chemotherapy regimens primarily target actively cycling "blast" cells and initially lead to remission. Since quiescent or protected LSCs often avoid eradication, relapse rates are high with 5-year survival rates below 15% for patients over the age of 60. A key goal is, therefore, to identify therapeutic strategies for targeting LSCs, while sparing healthy HSCs[9–11]. Characterizing gene expression differences between HSCs, pre-LSCs and LSCs would be a valuable step towards that goal.

Previously, LSC-specific gene expression patterns were characterized by isolating cells positive for stem cell-specific surface markers used in the healthy hematopoietic system, such as CD34 (refs.[12–15]). More recently, leukemic engraftment rates in xenotransplant models were correlated with gene expression[16–19]. However, these approaches all measure impure populations of cells. Here, we propose that by measuring mutational status and gene expression in single cells simultaneously, cancer stem cells can be uniquely distinguished from both mature cancer cells (based on gene expression) and healthy stem cells (based on mutational status) (Fig. 1a). Finally, pre-LSCs are thought to typically carry mutations associated with CHIP (e.g., in DNMT3A) but not mutations associated with leukemia (e.g., in NPM1)[7,20,21], potentially enabling their identification by profiling both known leukemic and known preleukemic mutations.

While we and others have demonstrated the utility of single-cell genomics for mapping hematopoietic differentiation hierarchies[22–24], tracking mutations or clones in single-cell gene expression data remains difficult. Previous work has amplified somatic mutations from complementary DNA (cDNA)[25–28], extracted mutational information from single-cell RNA-seq reads[29], or processed both genomic DNA and RNA from single cells[30–32]. However, these protocols suffer from a lack of confidence in assigning cells to clones and/or require prior knowledge of genomic mutation sites. As an alternative tool for clonal tracking, the use of endogenous mitochondrial mutations as clonal markers has been proposed, obviating the need for prior knowledge of genomic mutations[33,34]. However, the application of these methods to characterize LSCs has not been demonstrated, and in particular requires the ability to reliably detect clonal expansion events, associate clinically relevant coding mutations to clones with high confidence, and draw statements on gene expression changes between clones.

Here, we introduce MutaSeq, a workflow that amplifies nuclear mutations from cDNA, and mitoClone, a computational tool that achieves high-confidence clonal assignments and de novo discovery of clones using mitochondrial marker mutations when available. MutaSeq data from four AML patients allows us to distinguish HSCs, pre-LSCs, LSCs, and progenitor/blast populations. Thereby, we identify transcriptomic consequences of leukemic and pre-leukemic mutations relevant to stem cells. Additionally, we characterize the contribution of different leukemic and pre-leukemic clones to healthy and disease-specific bone marrow populations with unprecedented detail. Altogether, our results demonstrate cancer stem cell identification and characterization by simultaneous mapping of genomic and mitochondrial mutations in single-cell transcriptomes.

## Results

**MutaSeq provides high coverage of genomic and mitochondrial mutations.** To establish a robust experimental setup for the clonal tracking of human cells in single-cell transcriptomic data, we evaluated various modifications of the Smart-seq2 protocol aimed at increasing coverage at polymorphic genomic sites of interest (Supplementary Fig. 1a). We found that inclusion of targeting primers during reverse transcription frequently resulted in the formation of undesired byproducts, especially when targeting a higher number of sites (Supplementary Fig. 1a–d). By contrast, when sites of interest were targeted during cDNA amplification, we obtained high-quality transcriptome data while increasing the average number of target sites captured per cell by 2–4-fold compared to a non-targeted approach (Fig. 1b, c and Supplementary Fig. 1a, b). An automated pipeline for primer design that minimizes off-target sites and potential primer-dimer formation is available at https://github.com/veltenlab/PrimerDesign (see also the Methods section). MutaSeq stably works with up to 30–40 primer pairs (targets); when higher numbers of primers are included, library quality progressively decreases (Supplementary Fig. 1e). In a test with only highly expressed target genes, target amplicons are created from all primer pairs in virtually all single cells (Supplementary Fig. 1f, g).

To evaluate MutaSeq, we performed deep exome sequencing of an AML patient (P1) and designed primers targeting 14 nuclear mutations (Supplementary Data 1 and 4). We then systematically compared the performance of MutaSeq and non-targeted Smart-seq2 on CD34+ cells from this patient. MutaSeq increased the number of target sites covered per single cell from a median of 1 to a median of 4 (Fig. 1c–e) and recapitulated the variant allele frequencies estimated by exome sequencing with higher accuracy than Smart-seq2 (Fig. 1f), while maintaining comparable transcriptome data quality (Supplementary Fig. 1h, i). Both methods underestimated the abundance of frameshift mutations, possibly as a consequence of nonsense mediated decay[35] (Fig. 1e, f). While our data do not provide statistical evidence for an effect of target gene length or sequence complexity on dropout, we cannot exclude such effects.

Importantly, both methods provide an excellent coverage of the mitochondrial genome (mean ~100X mitochondrial coverage given a mean sequencing depth of ~788,000 total reads per cell, Fig. 1g), unlike most other single-cell RNA-seq protocols applied in the context of clonal tracking[27–29,31] (Supplementary Fig. 1j). Altogether, these results demonstrate that MutaSeq efficiently covers the mitochondrial genome in single-cell RNA-sequencing experiments and provides improved coverage of genomic target sites compared to Smart-seq2. Importantly, it requires no changes to existing Smart-seq2 pipelines, except for the addition of targeting primers during cDNA amplification.

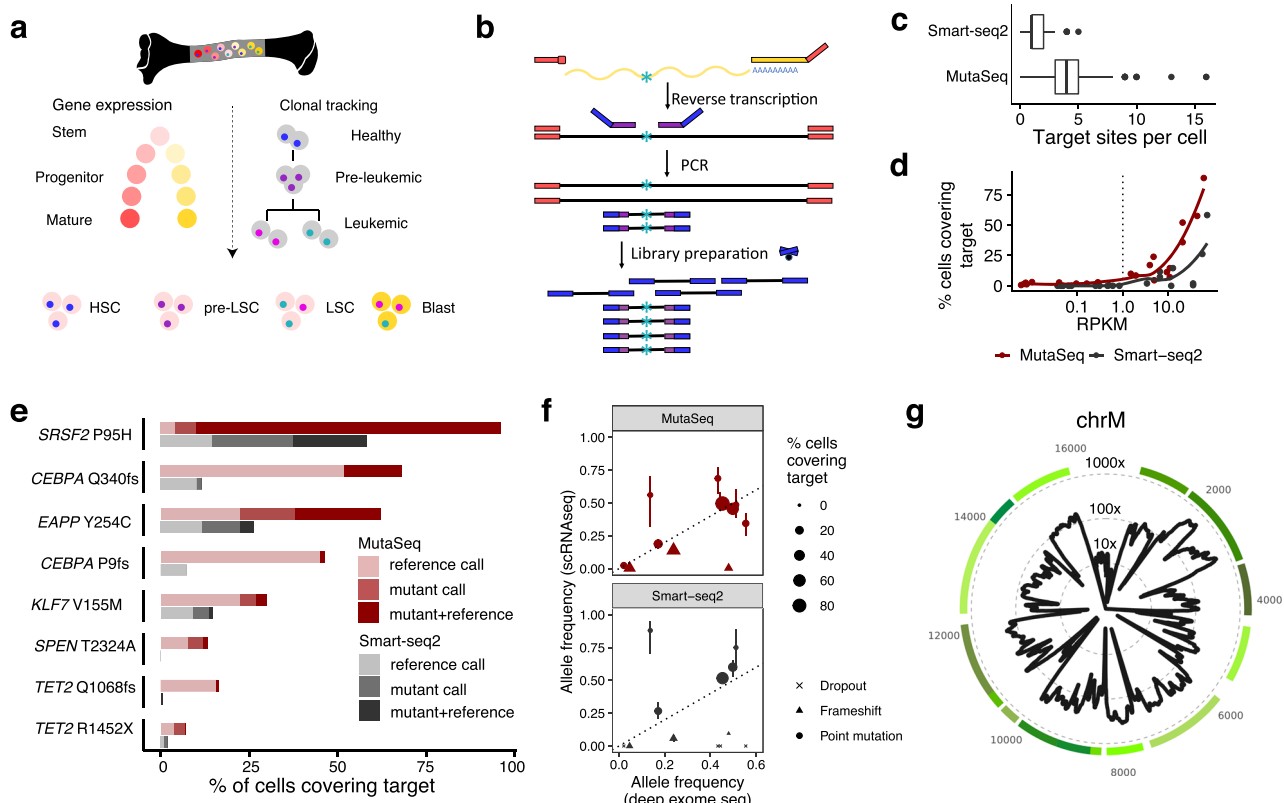

**Fig. 1 MutaSeq for high-quality single-cell RNA-seq data with clonal information.** See also Supplementary Fig. 1. **a** Overview of the study. HSC: hematopoietic stem cell, (pre-)LSC: (pre-)leukemic stem cell, Blast: mature leukemic blast. **b** Overview of the MutaSeq method. Targeting primers (purple) are included during the cDNA amplification step of the Smart-seq2 protocol. Targeting primers are directly fused to illumina library adapters (blue) and therefore get amplified efficiently during library preparation. Tagmentation introduces the same adapters to the full-length cDNA product. **c** Number of target sites covered per cell, across $n = 206$ (Smart-seq2) or $n = 658$ CD34+ (MutaSeq) bone marrow cells from patient P1 (see the Methods section Data visualization, for a definition of boxplot elements). **d** Mean gene expression of genes containing mutations of interest is plotted against the fraction of cells in which the mutation is covered. **e** Fractions of cells covering key non-synonymous mutations observed in the patient. Reference call: The reference allele was observed. Mutant call: the mutant allele, as defined by bulk exome sequencing (Supplementary Data 1) was observed. Mutant+Reference: both alleles were observed. **f** Allele frequency estimates derived from deep exome sequencing compared to allele frequency estimates derived from MutaSeq (red dots, $n = 2208$ single cells) or Smart-seq2 (gray dots, $n = 206$ single cells). Dot size indicates coverage at target site. Point shape indicates the type of mutation. Error bars indicates interquartile range. See figure source data for complete specification of sample size used to derive statistics ($n$). **g** Coverage of the mitochondrial genome in MutaSeq data. See Supplementary Fig. 1j for a comparison across methods. Green line segments correspond to genes in the mitochondrial genome.

**Simultaneous mapping of mitochondrial and genomic mutations permits high-confidence tracking of leukemic, pre-leukemic, and healthy clones.** To investigate if MutaSeq can distinguish leukemic, pre-leukemic and residual healthy clones, we generated data from four AML patients with heterogeneous genotypes and phenotypes (Figs. 2a, 3b and Supplementary Fig. 2). To allow for a better characterization of stem cells in each patient, cells were sorted such that putative stem and progenitor cells (CD34+) and putatively more mature cells (CD34-) were approximately covered at equal portions (Supplementary Fig. 2). Of note, two of the patients (P2, P4), exhibited bone marrow consisting of >99% of CD34− cells (Supplementary Fig. 2a). Owing to different capture rates across individuals and gates, the final data set consisted of between 618 to 1430 cells per patient, of which between 190 and 968 were CD34+ (Supplementary Fig. 2b, c). We therefore avoid statements on quantitative shifts in population size between patients throughout this manuscript.

We then called nuclear genomic mutations, as well as mitochondrial mutations, at the single-cell level in order to cluster cells into clonal hierarchies. Bulk exome sequencing of the patients had identified known pre-leukemic mutations present at high allele frequency and known leukemic mutations present at a

somewhat lower allele frequency (Fig. 2a and Supplementary Data 1. Patient 1: mutations in *SRSF2,TET2/CEBPA* and *SRSF2, TET2/KLF7*; Patient 2: mutations in *DNMT3A/NPM1*; Patient 3: mutations in *SRSF2/IDH2*; Patient 4: leukemic Trisomy 8 and *BRAF* mutations). While some statements on clonal hierarchies could be drawn solely based on calls of these nuclear somatic mutations (Supplementary Fig. 3a), the relatively high dropout of these sites impeded robust assignments of cells to clones (Supplementary Fig. 3b–d). Moreover, the result was biased by the expression levels of the mutated genes of interest: in cells with low expression, dropout was higher, leading to a higher fraction of false-negative calls, i.e., false classifications of mutant cells as reference (Fig. 1e and Supplementary Fig. 3c, d). Similar issues were faced by other methods using related approaches of mutation amplification from cDNA[27,28].

To overcome these limitations, we next determined if mitochondrial mutations can be used to refine clonal hierarchies jointly with the nuclear mutations. Since mitochondrial RNA is extensively edited, it is important to cluster cells based solely on mutations, and not on seemingly polymorphic sites that are the result of post-transcriptional events with no relationship to clonal structure. Here, we developed and tested a filtering strategy that

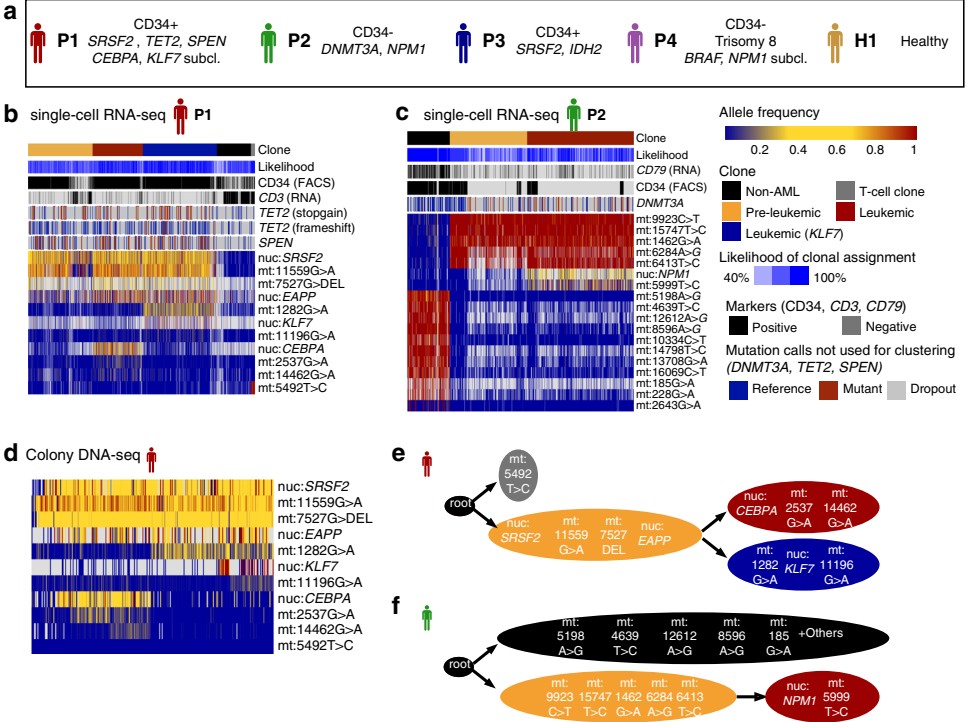

**Fig. 2 Mitochondrial mutations serve as high-confidence clonal markers in AML.** See also Supplementary Figs. 2–6. **a** Overview of the patients. CD34+ and CD34- indicate the dominant surface phenotype of the leukemic blasts (see Fig. 3b and Supplementary Fig. 2 for quantification). For each patient, genes containing mutations are printed in italic (see Supplementary Data 1 for a complete list). Subcl., sub-clonal mutation. **b** Heatmap depicting variant allele frequencies (color coded, see right of Fig. 2c for a key) observed in single-cell RNA-seq data of $n = 1430$ cells from P1. Gray indicates missing values. Cells and mutations are arranged according to the clustering result obtained by PhISCS[36] as described in the "Methods" section, *Analysis of mitochondrial mutations and reconstruction of clonal hierarchies*. Calculation of the likelihood is described in the same section. Mutations with low coverage (in *TET2*, and *SPEN*) were not included in the clustering and are depicted in the heatmaps as metadata, however, in all cases except *TET2* frameshift there is quantitative evidence for their association with specific clones (Supplementary Fig. 5b). For reproducing the computations, see the vignettes accompanying the mitoClone package. For mutations, nuc is nuclear genome; mt is mitochondrial genome. **c** Like panel **b**, but using $n = 1066$ cells from P2. DNMT3A is included as a low coverage mutation with significant association to the pre-leukemic and leukemic clone (Supplementary Fig. 5b). **d** Heatmap depicting variant allele frequencies (color coded, see legend to the right of Fig. 2c) observed in targeted DNA-seq data from $n = 288$ single-cell derived colonies from P1. **e** PhISCS[36] was run on the mutational data from P1 to reconstruct a clonal hierarchy, see the Methods section Analysis of mitochondrial mutations and reconstruction of clonal hierarchies. Take note that while the order of mutations is based on the PhISCS model, the grouping of mutations into clones is based on an arbitrary cutoff to provide a useful clustering for further analyses. See Clone in Fig. 2c legend for color codes. **f** Like **e**, but for P2.

only makes use of mitochondrial mutations for clonal tracking if they uniquely occur in individual patients (Supplementary Fig. 4a, see also Methods). This idea assumes that RNA editing events are typically shared between individuals, whereas somatic mutations are not. Using the whole-exome sequencing data, we validated that this approach, at the level of genomic sites, correctly distinguishes mutations and RNA editing events with a precision of 97% (Supplementary Fig. 4b, c). We further analyzed various control data sets with known associations between mitochondrial mutations and clones[33] to validate that this approach enables the detection of relevant mitochondrial mutations without a need for a DNA-based reference, and further enables the unsupervised identification of clones (Supplementary Fig. 4d–f). We have implemented all the required filtering and blacklisting routines required for the identification of high-confidence somatic mitochondrial variants in the mitoClone R package (https://github.com/veltenlab/mitoClone).

We then computed clonal hierarchies from both nuclear and mitochondrial somatic variants using a mathematical model that accounts for allelic dropout[36] (see Methods for detail, all required tools are contained in the mitoClone package). In patients P1 and P2, pre-leukemic as well as sub-clonal leukemic mutations were significantly associated with distinct sets of well-covered mito-chondrial variants (Supplementary Fig. 5a, b), such that clonal

hierarchies could be delineated, and a confident assignment of cells to clones became possible (Fig. 2b, c and see Supplementary Fig. 3b for a quantitative analysis). Unlike genomic mutation calling from cDNA, identification of clonal identities from mitochondrial mutations is mostly not, and in one case weakly, affected by gene expression levels or library quality (Supplementary Figs. 3c, d, 5c), and is possible at lower sequencing depths (Supplementary Fig. 5d–f), since mitochondrial genes are consistently highly expressed.

Importantly, the clonal structure was validated by targeted DNA-sequencing from single-cell derived colonies (Patient 1, Fig. 2d). Across the patients, we identified clones carrying known pre-leukemic mutations (e.g., in *SRSF2*, *DNMT3A*) and sub-clones carrying known leukemic mutations (e.g., *CEBPA*, *NPM1*) (Fig. 2e, f). Below we functionally characterize these clones as leukemic or pre-leukemic based on their contribution to healthy blood production (Fig. 4).

In patient P3, allele frequencies of leukemic and pre-leukemic mutations were near 50%, indicating the presence of a single leukemic clone (Supplementary Data 1). In patient P4, an 18-year old individual with a leukemia driven by triplication of chromosome 8, no mitochondrial markers were identified, even though exome sequencing suggested the presence of sub-clonal variants (Supplementary Data 1). In this case, MutaSeq still

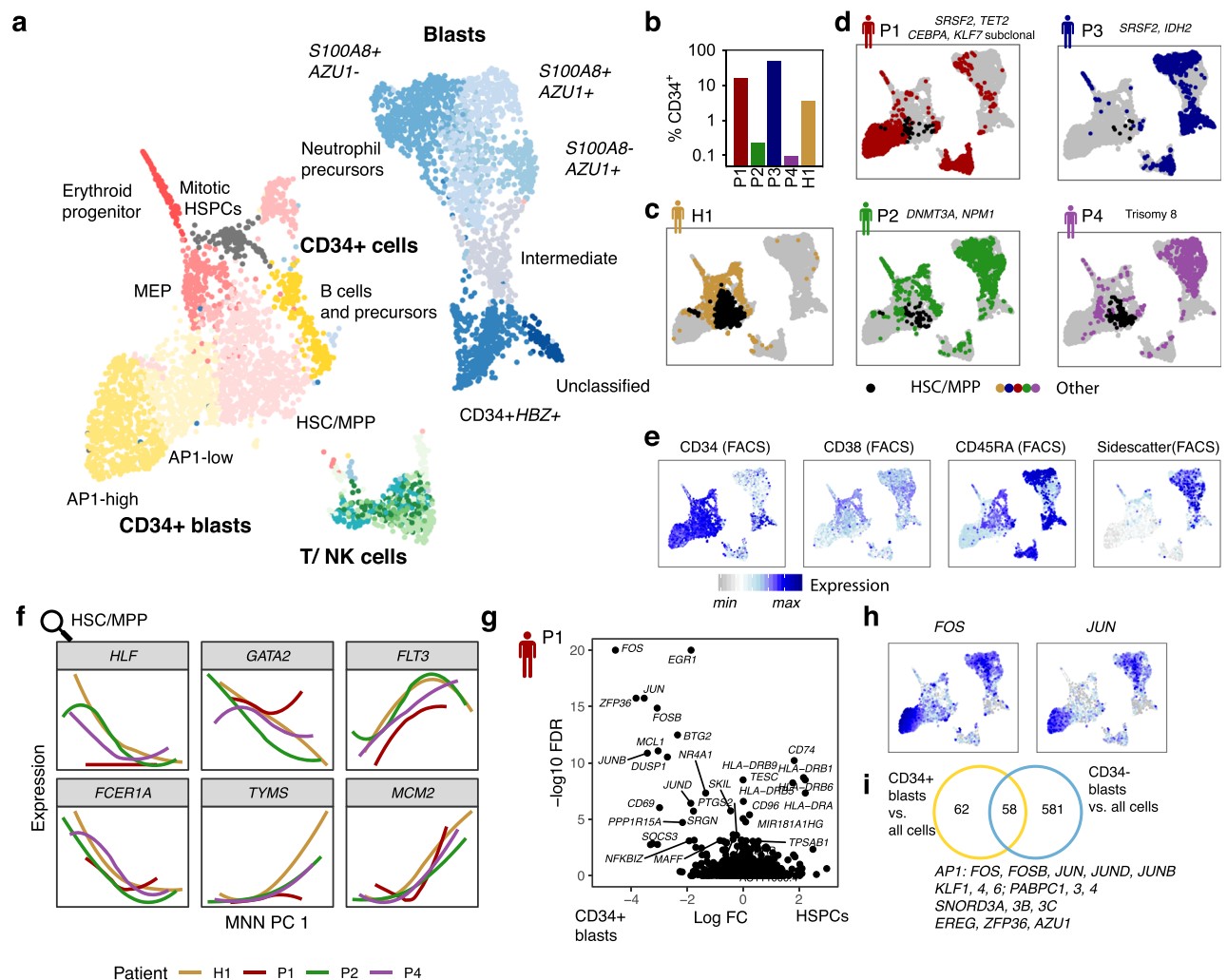

**Fig. 3 An overview of cell types observed in bone marrow of healthy and leukemic individuals.** See also Supplementary Figs. 2 and 7. **a** Data from the five individuals (Fig. 1a) were integrated using scanorama[60] and visualized in two dimensions using uMAP[62,63]. Clusters are color-coded. N = 5228 cells. **b** Fraction of CD34+ cells in total bone marrow from the five individuals. CD34+ cells were enriched during FACS sorting for a higher representation in the single-cell RNA-seq data (Supplementary Fig. 2). **c** CD34+ cells from a healthy individual[22] highlighted on the uMAP. Black dots correspond to cells from the HSC/MPP cluster. **d** Cells from each patient were highlighted separately on the uMAP. Black dots correspond to cells from the HSC/MPP cluster. **e** Logicle-transformed expression of key FACS markers highlighted on the uMAP (Supplementary Fig. 7a). **f** Data from n = 667 cells from the HSC/MPP cluster were integrated across individuals using MNN[64]. The smoothened expression of several marker genes of healthy HSC/MPP subsets[22] is plotted over the first dimension of variability identified by MNN (Supplementary Fig. 7f). **g** Volcano plot of the log10 expression change (Log FC) in n = 569 AP1-high CD34+ blasts vs. n = 667 HSC/MPP-like cells, plotted against corrected p-values from MAST, using a model that accounts for differences in library quality and patient identity/batch (see the Methods section Single-cell gene expression data analysis). AP1-high CD34 + blasts were chosen for this comparison since AP1-low blasts, in terms of all marker genes, appear to constitute an intermediate state between Healthy-like HSC/MPPs and AP1-high blasts. **h** Log-normalized expression of *FOS* and *JUN* on the uMAP from panel **a**. See panel **e** for a color scale. **i** Venn diagram displaying genes with significant overexpression in AP1-high CD34+ blasts and CD34− blasts, compared to all other cells from the data set.

permits a qualitative analysis using genomic mutations alone (see below). Taken together, our approach allows for the identification of putatively leukemic, pre-leukemic, and healthy clones and can assign cells to clones with high confidence if mitochondrial somatic variation is present.

**Identification and characterization of clones de novo.** We next investigated if the use of mitochondrial somatic variants enables the identification and characterization of clones without prior knowledge of nuclear mutations. To that end, we made use of a data set from patient P1 generated without amplification of nuclear sites (i.e., standard Smart-seq2). A clear clonal structure was identified in an unsupervised manner based solely on

mitochondrial variants (Supplementary Fig. 6a). In order to examine whether the presence of somatic genetic variability is associated with the different clones, we then queried the mutational status of 13,797 genomic sites annotated as mutated in AML in the COSMIC database[37] using a beta-binomial model (see Methods). This unsupervised analysis revealed a highly significant association of the *SRSF2* P95H mutation with the leukemic and pre-leukemic sub-clones (Supplementary Fig. 6b, c). Their malignant nature was further evidenced by a markedly reduced ability to contribute to the T cell lineage (Supplementary Fig. 6d and see also below).

To further demonstrate our ability to identify clones de novo, we highlight a clonal expansion of non-leukemic cells in P2 (Fig. 2f).

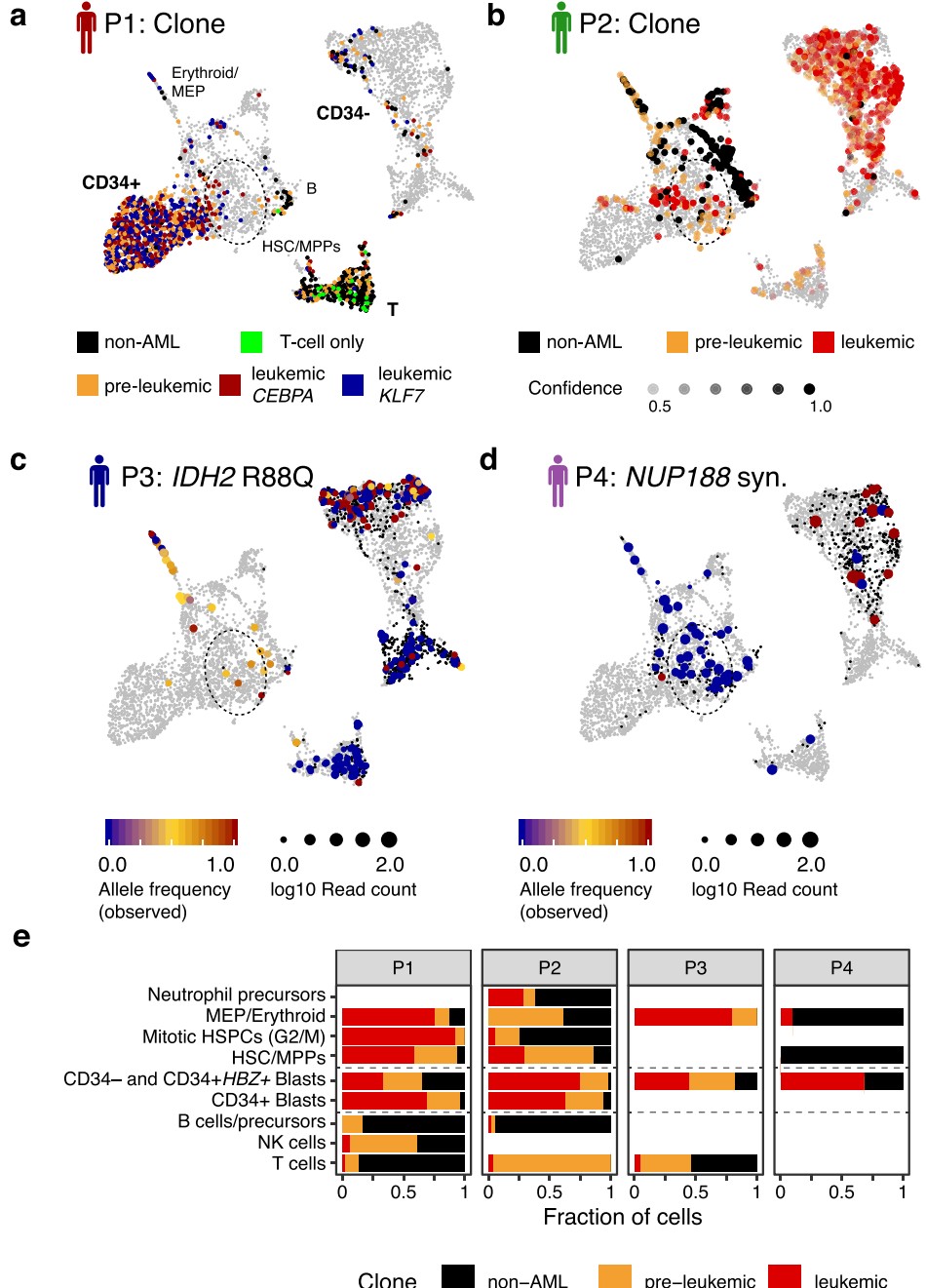

**Fig. 4 Effects of (pre-)leukemic mutations on cellular differentiation.** See also Supplementary Fig. 8. **a** Clonal identity of the cells from P1 highlighted on the uMAP (Figs. 2b, e, 3a). Gray dots correspond to cells from other patients. The dotted ellipse serves as a guide to identify the location of the HSC/MPP population. **b** Clonal identity of the cells from P2 highlighted on the uMAP (Figs. 2c, f, 3a). **c** Observed variant allele frequencies for the *IDH2* R88Q mutation from P3 highlighted on the uMAP. Small black dots correspond to cells with no coverage of the mutation (see Supplementary Fig. 8a for an estimate of target capture rates). **d** Observed variant allele frequencies for the synonymous *NUP188* mutation from P4 highlighted on the uMAP. This mutation was observed at an allele frequency of 50% in exome data of total bone marrow. Note that CD34+ cells were enriched more than 100-fold during sorting for single-cell RNA-seq (Supplementary Fig. 2 and see Supplementary Fig. 8a for an estimate of target capture rates). **e** Estimate of the contribution of different clones to the cell types. For P1 and P2, clonal identities from Fig. 2 were used. For P3, cell were classified as leukemic if the *IDH2* mutation was observed, as pre-leukemic if the *SRSF2* mutation was observed, or as non-leukemic if the reference allele was observed for both mutations. Cells without coverage were excluded from the analysis. For P4, cells were classified as leukemic if the *NUP188* mutation was observed, or as non-leukemic if the reference allele was observed. Bars are only shown for populations covered with at least 10 cells (see Supplementary Fig. 8b for absolute numbers and Supplementary Fig. 8c, d for a quantitative analysis).

This clone would have been missed by approaches relying on genomic mutations alone[27,28,31]. Interestingly, these cells were not associated with the pre-leukemic *DNMT3A* mutation. By again querying sites from the COSMIC database[37] using a beta-binomial model, we identified that they had uniquely acquired a mutation in the *RPL3* gene (Supplementary Fig. 6b, e). These results suggest that this clonal expansion event is independent of the leukemia and associated with the acquisition of unrelated nuclear mutations.

We also take note of a putative non-leukemic clone in P1 marked by a single mitochondrial variant (5492T>C). With one exception, all cells carrying this variant are positive for the T-cell marker CD3 (Figs. 2b, 4a). Hence, this variant was likely acquired in a T-cell precursor or T-cell clone, although we cannot formally exclude that it corresponds to a T-cell-specific RNA editing event.

Taken together, these results demonstrate that our approach can identify and characterize clones de novo without prior knowledge of nuclear genomic mutations. The mitoClone package implements all routines for clonal clustering and mutation calling.

**A map of cell states in leukemic bone marrow identifies HSC-like cells.** We next used single-cell transcriptome data to distinguish stem cells, progenitor cells and leukemic blasts. To define cell populations in our samples, we integrated the gene expression data from all four patients along with data from CD34+ bone marrow cells of a healthy individual[22] into a two-dimensional representation, and characterized cell types based on marker gene expression (Fig. 3a, Supplementary Fig. 7a–c, f, Supplementary Data 2).

All four bone marrow samples contained cells that clustered with HSCs and multipotent progenitor cells (HSC/MPP) from the healthy individual (Fig. 3b–d) and displayed a CD34+CD38low, FSC/SSClow phenotype (Fig. 3e). Unsupervised analysis separated these cells into quiescent immature HSC-like cells (HLF+), more proliferative erythromyeloid primed progenitors (GATA2+), and lymphomyeloid primed progenitors (FLT3+) (Fig. 3f and Supplementary Fig. 7f). Cells resembling healthy erythroid progenitors and MEPs were also identified, alongside various types of B-cell precursors and T/NK cells (Supplementary Fig. 7b, c).

In contrast to non-leukemic populations, the majority of cells from leukemic bone marrow samples ("blasts") were very different between the patients, in line with previous observations[27] (Fig. 3d): we observed (a) differentiated blasts (CD34-SSC/FSChi) that expressed neutrophil genes such as calprotectin (S100A8, S100A9) and AZU1 (Fig. 3e and Supplementary Fig. 7a); (b) CD34+ blasts that were highly mitotic and expressed the fetal hemoglobin HBZ; and c) CD34+CD38low blasts expressing markers typical of hematopoietic stem and progenitor cells (HSPCs), such as PROM1 (CD133) and MEIS1 (Supplementary Fig. 7a, d, e). The latter population appeared to be connected to the HSC/MPP population across a continuum of states that gradually upregulate AP1 transcription factor expression (FOS, JUN, FOSB, JUNB, JUND) while down-regulating MHC class II (Fig. 3g, h and Supplementary Data 3). A global analysis of highly expressed genes across all populations revealed that high expression levels of AP1, as well as several Krüppel-like factors and poly-A binding proteins, were common to all different blast populations from the patients and distinguished them from healthy progenitors (Fig. 3h, i). Of note, high AP1 and KLF activity have recently been identified as a hallmark conserved across genetically distinct types of AML by bulk-sequencing studies that only investigated blast populations[38]. Our results demonstrate that indeed these transcription factors appear to be relevant in all, phenotypically very different, blast populations.

Taken together, all four leukemia samples could be stratified into stem cells, progenitors, and blasts. Furthermore, all patients retain cells highly similar to healthy HSCs, although in variable abundance.

**Clonal tracking identifies cellular differentiation states and gene expression patterns associated with pre-leukemic and leukemic mutations.** Gene expression information alone was insufficient to distinguish cancerous from non-cancerous HSPCs

(Supplementary Fig. 7f). To definitively characterize cells as (pre-)LSCs or residual healthy cells, we therefore integrated single-cell gene expression data and clonal tracking results (Fig. 4–d). If mitochondrial somatic variability was present, we were able to assign clonal identities with high confidence, allowing us to draw quantitative statements (Patients P1 and P2). In the absence of mitochondrial somatic variability, we used nuclear mutation calls in SRSF2, IDH2 and NUP188 for purely qualitative statements (Patients P3 and P4). The capture rates of these marker sites ranged from 70% (SRSF2) to 11% (NUP188) (Supplementary Fig. 8a).

Clones associated with leukemic mutations were most prevalent in the blast compartments and were also detected in the HSPC compartment, but were almost absent in lymphoid (B, NK, T) lineages. By contrast, clones associated with pre-leukemic mutations were found in all lineages, but mostly displayed a decreased prevalence in lymphoid lineages (Fig. 4e and Supplementary Fig. 8b–d). These observations confirm the designation of these clones as "leukemic" and "pre-leukemic". Furthermore, these results highlight that the leukemic mutations may initiate differentiation blocks at various levels, as previously reported[4,27,39]. For example, in patients P1 and P3, leukemic cells had retained the ability to contribute to the erythroid lineage, while in patient P2, this activity was restricted to the pre-leukemic and non-leukemic clones. Importantly, the leukemic cells in P1, P2, and P3 were observed in a cell state that is highly reminiscent of healthy HSCs/MPPs on a molecular level, and retains the ability to contribute to various lineages, i.e., is functionally multipotent.

Next, we investigated the molecular effects of distinct mutations in detail. Previously, the consequences of specific leukemic mutations were commonly studied in mouse models. MutaSeq data allows us to compare gene expression between clones differing only in a single mutation, thereby elucidating the specific effects of that mutation on human hematopoiesis.

Mutations in the de novo DNA methyl transferase DNMT3A are the most common cause of benign clonal expansions of HSCs in individuals of advanced age[6,7]. In patient P2, DNMT3A-mutated pre-leukemic HSPCs were rather stem-like (with relatively high expression of HLF) or primed into the erythro-myeloid direction (with relatively high expression of GATA2) (Figs. 4b and 5a). These results are in line with recent findings from DNMT3A knock-out mouse models[40]. Interestingly, independent of cell state, this coincided with an upregulation of MLLT3, a gene whose enforced expression promotes erythroid-megakaryocytic output from HSCs[41] (Fig. 5b, c and Supplementary Data 3).

Mutations in the multifunctional ribonucleoprotein NPM1 are identified as drivers for acute myeloid leukemia in 30% of patients and frequently co-occur with pre-leukemic DNMT3A mutations[21]. In patient P2, the erythromyeloid bias of the DNMT3A clone was lost upon acquiring the leukemic NPM1 mutation. NPM1 mutated cells upregulated HOXB3, again in line with data from mouse models[42–44] (Fig. 5b, c). Independent of cell state, these cells further exhibited upregulation of CD96 RNA expression, which has previously been identified as a leukemia stem cell-specific marker[12]. CD96 was also highly expressed on leukemic HSC/MPP-like cells of patient P3, but not in patient P1, further illustrating the patient-specific nature of LSC markers (Supplementary Fig. 8e).

Mutations in KLF7 are not commonly observed in leukemia; however, krüppel-like factors are highly expressed by genetically and phenotypically different blasts[38] (see above). In patient P1, the KLF7 mutated clone displayed a higher proportion of cells in G2/M phase (Fig. 5d). Based on a reanalysis of ATAC-seq data from human CD34+ cells[45] we found that enhancers containing KLF7-binding sites were enriched near tumor suppressor genes[46],

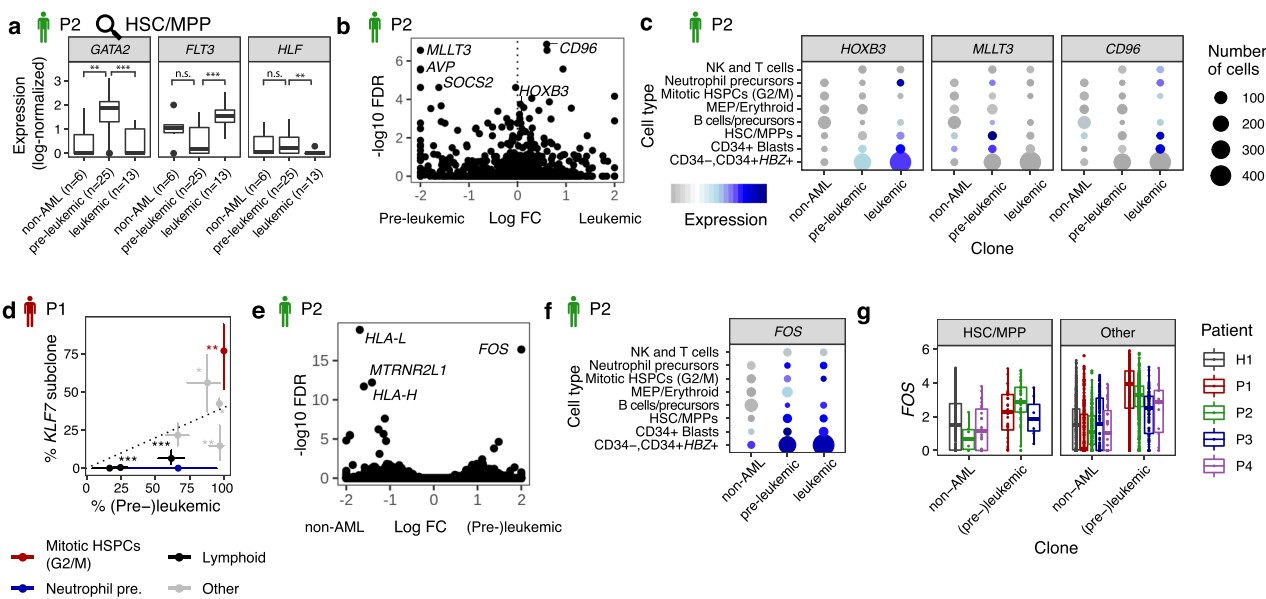

**Fig. 5 Molecular consequences of leukemic and pre-leukemic mutations.** See also Supplementary Fig. 8. **a** Log-normalized expression of *GATA2*, *FLT3* and *HLF* in HSC/MPPs from P2, stratified by clonal identity (see the Methods section Single-cell gene expression data analysis, for detail on data normalization). Asterisk indicate significance from a two-sided Wilcoxon test, ***$p < 0.001$, **$p < 0.01$, n.s.: not significant (see the Methods section Data Visualization, for a definition of boxplot elements). **b** Volcano plot of the log10 expression change (log FC) in pre-leukemic ($n = 55$) vs. leukemic ($n = 50$) CD34 + cells of P2, plotted against corrected *p*-values (FDR) from MAST[65], using a model that accounts for differences in cell type and library quality (see Methods). Only the following CD34+ cell types from Fig. 3a were included in the test: HSC/MPP, CD34 + Blasts (both subsets), Neutrophil precursors, and MEPs. **c** Dot plot comparing the expression of relevant genes across non-leukemic, pre-leukemic, and leukemic cells in the different cell types. Symbol size scales with the number of cells per cell type. **d** Scatter plot depicting the fraction of (pre)leukemic cells in relation to the fraction of cells from the *KLF7*-mutated clone in various cell types in P1. Dotted line indicates the mean ratio across all cells, error bars denote 95% confidence intervals from a beta distribution, and asterisk indicate significant deviation from the mean ratio, as follows: *$p < 0.05$; **$p < 0.01$; ***$p < 0.001$. *p*-values are from a two-sided binomial test and were not adjusted for multiple testing (see Supplementary Fig. 8c and figure source data for number of single cells underlying each group). **e** Volcano plot of the log10 expression change (log FC) in (pre-)leukemic ($n = 105$) vs. non-leukemic ($n = 41$) CD34 + cells of P3, plotted against corrected p-values (FDR) from MAST[65], using a model that accounts for differences in cell type and library quality (see Methods). Only the following cell types from Fig. 3a were included in the test: HSC/MPP-like, CD34+ Blasts (AP1 high and low), Neutrophil precursors, and MEPs. **f** Dot plot comparing the expression of *FOS* across non-leukemic, pre-leukemic and leukemic cells in the different cell types. Dot size and color represent the quantity of cells and gene expression level, respectively (see legend in Fig. 5c). **g** Boxplots comparing the log-normalized expression levels of *FOS* between cells with evidence of originating from the non-leukemic clone(s), and cells with evidence of originating from the leukemic or pre-leukemic clones. Cells were assigned to clones as in Fig. 4e (see the Methods section Single-cell gene expression data analysis, for detail on data normalization, see figure source data for number of single cells underlying each group and see the Methods section Data Visualization, for a definition of boxplot elements).

including *CDK6*, *PTEN*, *RUNX1*, and *FLT3*, compared to active enhancers not containing KLF7 binding sites ($p = 0.002$, Supplementary Fig. 8f).

In sum, we have used a small heterogeneous patient cohort to demonstrate, as a proof-of-concept, that the acquisition of specific mutations is frequently linked to an altered gene expression program, which is consistent with data obtained from mouse models[40,42–44].

Finally, we compared gene expression between all (pre-)leukemic and non-leukemic cells, with the goal of identifying potential markers or drug targets present in all (pre-)leukemic cells, but not in residual healthy cells. In patient P2, our analysis identified a small number of hits (Fig. 5e and Supplementary Data 3), most notably *FOS*, which was expressed in all (pre-)leukemic cells across cell types, but not by non-leukemic HSC/MPPs (Fig. 5f), and the MHC class I genes *HLA-L* and *HLA-H*, which were expressed by all healthy cells, but not by (pre-)leukemic cells. Across all patients, *FOS* was consistently over-expressed in cells carrying (pre-)leukemic lesions, both in HSCs/MPPs, and in other cell types (Fig. 5g).

Taken together, these results demonstrate the ability of MutaSeq and mitoClone to delineate developmental and molecular effects of clonal evolution caused by leukemic and pre-leukemic mutations.

## Discussion

Herein, we have described a joint single-cell transcriptomics and clonal tracking approach (MutaSeq and mitoClone) for characterizing LSCs, charting their differentiation capabilities, and mapping the molecular consequences of oncogenic mutations. While single-cell gene expression profiling permits the identification of cells with a stem-cell signature, clonal tracking using genomic and mitochondrial mutations allows for a clean separation between healthy and cancerous clones. Thereby we distinguish LSCs, HSCs, pre-LSCs, healthy progenitors, and blasts. We have demonstrated this approach in the context of acute myeloid leukemia, and we propose that similar approaches may be applied to other types of cancers.

By applying our approach to bone marrow samples from four AML patients, we have demonstrated its capabilities:

*Charting the differentiation capacities of hematopoietic clones.* By quantifying the contribution of (pre-)leukemic cells to blood lineages, we have shown that in patients P1–P3, leukemic clones not only form blasts but also exist in an HSC-like state. Pre-leukemic clones additionally contribute to erythroid and lymphoid lineages. Multipotent HSCs are, therefore, the likely cell of origin in these patients[4]. In patient P4, with one exception, only cells with a mature phenotype displayed leukemic mutations, illustrating that the disease can also be fueled by cells with a

committed phenotype[39]. Alternatively, the LSC population in this patient might be rare among CD34+ cells.

*Identification of clones de novo.* In the presence of mitochondrial somatic variability, our approach does not rely on previously known nuclear mutations to detect clones. In the example of patient 2, we have thereby identified an expanded clone present at very low frequency in total bone marrow, but highly abundant in the CD34+ fraction. We have demonstrated that this clone is associated with a nuclear mutation in the *RPL3* gene, which we discovered de novo. This result illustrates the low clonal complexity of hematopoiesis in individuals of advanced age, which is often not associated with candidate driver mutations[20].

*Identification and characterization of LSCs.* Importantly, our approach was capable of identifying leukemic cells highly reminiscent of healthy HSCs. These cells will be important to target therapeutically without ablating their healthy counterparts[9–11]. While stemming from a study cohort of limited size, our data suggest that *FOS* might constitute a potential target for further investigation, as it is expressed throughout the disparate leukemic populations including the most HSC-like cells, pre-LSCs, and blasts, but not in healthy HSCs. Further, we have confirmed that CD96 is a specific marker for LSCs in some patients[12]. Studies in larger cohorts will be required to assess how generally applicable these findings are.

Taken together, our study expands upon earlier work on the molecular phenotypes of AML blasts[27,38] and constitutes the first detailed characterization of LSCs by single-cell transcriptomics. This advance is owed to three crucial aspects of experimental design.

*Enrichment of relevant starting populations.* LSCs are excessively rare and generally present at below 0.1% of total bone marrow[16]. In order to characterize these cells by single-cell genomics, a prior enrichment, e.g., by sorting for CD34 expression, is essential. In future studies, this approach can be complemented with markers that also label CD34- LSCs, such as GPR56 (ref. [18]).

*Deep transcriptome sequencing.* In some cases, the bulk of leukemic cells displays gene expression signatures highly similar to stem cells, as observed here for the CD34+ blasts of patient P1. The differences between LSCs and residual healthy HSCs are even more subtle. Previous work using shallow, microwell-based sequencing of AML cells[27] has not identified differences between LSCs, CD34+ blasts and residual healthy HSCs.

*High-confidence clonal tracking.* When available, the use of mitochondrial variants enables the confident assignment of cells to clones, and thereby, a quantitative analysis of gene expression. In two out of four AML patients, we identified high-confidence, specific mitochondrial genetic markers for pre-leukemic and leukemic sub-clones. In the third patient, allele frequencies of leukemic and pre-leukemic mutations were around 50%, indicating dominance of a single leukemic clone. In the final patient, sub-clonal genomic mutations were observed, but not associated with mitochondrial variability. Interestingly, this patient was only 18 years old and exhibited a leukemia possibly driven by a "catastrophic" triplication of chromosome 8. The length of the pre-leukemic phase, the buildup of mitochondrial variants accumulated during normal ageing, as well as unknown factors affecting mitochondrial mutation rates might all contribute to the presence of mitochondrial marker mutations.

In the context of recent methods for clonal tracking within single-cell transcriptomics[27–29,31,33], the use of the MutaSeq protocol and mitoClone computational pipeline combines the strengths of previous approaches relying on either nuclear or mitochondrial variants, but also has limitations. Specifically, droplet- or microwell-based protocols for single-cell RNA-seq and clonal tracking[27–29] provide low coverage of mitochondrial

genomes and high dropout of nuclear genomic sites, and therefore do not enable a confident assignment of cells to clones. Single-cell ATAC-sequencing[33,34,47] and Smart-seq2 efficiently cover mitochondrial genomes, but their coverage of nuclear genomic sites is absent or low, hardening interpretability of the data. Finally, while TARGET-seq addresses the limitation of dropout of nuclear sites, depending on the implementation it does not offer sufficient mitochondrial coverage, or is of very limited throughput. Of all methods available to date, the approach presented here converges on crucial aspects, specifically: (a) high mitochondrial coverage, allowing us to identify benign expanded clones de novo, even in the absence of known genetic markers, (b) decreased dropout of relevant genomic mutations, permitting the association of clones with genomic mutations, if present, and (c) a highly confident assignment of cells to clones, enabling quantitative analyses of clone-specific gene expression (Supplementary Fig. 9). These capabilities expand the potential applications of our approach to the study of clonal dynamics during ageing and oncogenesis beyond the hematopoietic system.

**Limitations.** The major limitation of our pipeline is that it requires natural somatic variability to resolve clones at high confidence. In the absence of mitochondrial somatic variation, the MutaSeq protocol can be used to draw qualitative statements on clonal differentiation capacities (similar to ref. [28], and with an improvement over Smart-seq2), but due to dropout neither enables the statistically confident assignment of cells to clones, nor differential expression testing between clones. In the presence of mitochondrial somatic variation, an association between clones and nuclear mutations is only possible for mutations in highly expressed genes, and can be limited by the nature of the mutation (e.g., frameshift mutations) and possibly other factors such as sequence complexity. The third limitation of MutaSeq is its relatively low throughput. This limitation is currently shared with Smart-seq2 and alternative single-cell RNA-seq methods allowing high-confident assignment of cells to clones[31,33]. Future work will focus on the inclusion of full-length coverage of the mitochondrial genome in droplet-based single-cell RNA-seq platforms.

## Methods

**Patient and sample collection.** The AML samples were collected from diagnostic bone marrow aspirations at the University hospitals in Heidelberg, Germany, and Mannheim, Germany after obtaining written consent. Bone marrow mononuclear cells were isolated by density gradient centrifugation and stored in liquid nitrogen until further use. All experiments involving human samples were conducted in compliance with the Declaration of Helsinki and all relevant ethical regulations and were approved by the ethics committees of the medical faculties Heidelberg and Heidelberg-Mannheim of the University of Heidelberg, the Bioethics Internal Advisory Committee (BIAC) at EMBL and the CRG bioethics committee (CEIC-Parc de Salut Mar).

**Deep exome sequencing and target selection.** For exome sequencing, DNA was extracted from $9 \times 10^3$ flow sorted CD34 + cells (for CD34+ leukemias: P1, P3) or total bone marrow. As healthy controls, we used a buccal swap (P1), FACS-sorted CD45-CD105+MSCs (P3) or in vitro expanded MSCs (P2 and P4)[48]. Sequencing libraries were constructed using the SureSelect HS XT Target Enrichment System v6 (Agilent), and a mean on-exon sequencing coverage of at least >70X was obtained for each patient. Genomic alignments were performed using BWA MEM v0.7.15 (ref. [49]) and cancer variants were identified using Mutect2 v3.8 (P1 and P3) and v4.0.9 (P2 and P4)[50], following the GATK best practice recommendations. Variants were annotated using ANNOVAR[51]. Output from Mutect2 was filtered to remove variants that did not overlap with known genes. The final list of candidate variants included only those with allele frequencies (AF) >4% in the cancer exome sample and with an AF fourfold larger than in the healthy exome sample (Supplementary Data 1). Finally, the candidates for targeting were hand-selected from this list with a focus on cancer relevant genes, highly expressed genes, and potential sub-clonal markers.

**FACS sorting.** Bone marrow mononuclear cells were stained according to standard protocols. In brief, cells were thawed, washed once in medium (IMDM, 10% FCS,

20 U/mL DNAse I) and 3 million cells were resuspended in 100 μL medium containing antibodies diluted as described in Supplementary Table 1. Following incubation for 30 min on ice, cells were resuspended in phosphate-buffered saline with 2% FCS. For single-cell liquid cultures and MutaSeq, cells were stained with fluorescent-labeled antibodies against lineage markers (CD4, CD8, CD19, CD20, CD41a, CD235a) and additional markers (CD45RA, CD135, GPR56, CD34, CD38, CD90, CD33, Tim3), and sorted according to the gating scheme illustrated in Supplementary Fig. 2. BD FACS Fusion (BD Biosciences) equipped with 405, 488, 561, and 640 nm lasers were used. Of note, in P1 and P4, Lin+ cells could not be efficiently processed into libraries for unknown reasons and are, therefore, excluded. A list of all antibodies used can be found in Supplementary Table 1.

**Cell culture**. K562 cells were purchased from ATCC (catalog number CCL-243) and cultivated in RPMI-1640 (Thermo 21875034) supplemented with 10% FBS and P/S.

**Primer design**. Primers for MutaSeq, for other single-cell targeting protocols tested (Supplementary Fig. 1), as well as for targeted DNA sequencing were designed using the computational pipeline available at http://git.embl.de/velten/PrimerDesign. For MutaSeq, the refgene transcripts spanning each genomic site of interest were selected as template; if multiple refgene transcripts were found for one site, a consensus transcript containing only exonic sequences present in all variants was created. We then used primer3 (ref. [52]) to design five possible pairs of primers for each intended target, with an amplicon length of 90–145 bp and a melting temperature of (nominally) 60 °C. BLAST was used to remove primer pairs, which potentially form off-target amplicons. Then, the pair complementarity (i.e., potential to form dimers) was computed for each possible combination of primers across all target sites (forward-reverse, forward-forward and reverse-reverse). In order to identify a set of primers that covered the maximal number of genes while strictly forbidding primers with high complementarity scores, a graph was constructed that connected all primers with different targets to each other if their complementarity score was lower than 15. A maximum clique-finding algorithm[53] was then used to identify the largest mutually connected component in the graph. Thereby, the largest number of targets that efficiently avoids dimer formation was selected.

Nextera adapters were added to all primers designed accordingly (fwd: GTCGT CGGCAGCGTCAGATGTGTATAAGAGACAG, rev: GTCTCGTGGGCTCGGA GATGTGTATAAGAGACAG).

For targeted DNA-seq experiments, the genomic sequence surrounding the target was used as template and nested PCR primers were designed. Inner primers were designed as in the case of MutaSeq, and outer primers surrounding the inner PCR product with an amplicon length of 200–350 bp and a nominal annealing temperature of 58 °C were added. A list of all primers used for this study is included in Supplementary Data 4.

**Single-cell RNA sequencing with targeting of genomic sites of interest (MutaSeq)**. MutaSeq is based on the Smart-seq2 protocol[54,55] with the modifications introduced by ref. [22]. For lysis, we used 5 μL of a buffer containing 0.1 μL RNAsin + (Promega), 0.04 μL 10% Triton X-100 (SigmaAldrich), 0.1 μL of 100 μM Smart-seq2 Oligo-dT primer (SigmaAldrich) and 1 μL dNTP mix (10 mM each, NEB). In P1, we had additionally included 0.075 μL of a 1:1,000,000 dilution of ERCC spike-in mix 1 (Ambion), as well as a control spike-in to quantify the false-positive detection rate of mutations (Supplementary Fig. 10). Plates were snap frozen directly after sorting and later thawed at 10 °C in a PCR machine for 5 min and denatured at 72 °C for 3 min. 5 μL of a buffer containing 0.25 μL RNAsin +, 2 μL 5x SMART FS buffer, 0.5 μL DTT 20 mM, 1 μL SmartScribe enzyme (all TaKaRa) and 0.2 μL 50 μM Smart-seq2 TSO (Exiqon) were then added and RT was performed for 90 min at 42 °C, 10 cycles of [50 °C, 2 min and 42 °C, 2 min], and enzyme inactivation at 70 °C for 15 min. Then, we added 15 μL PCR mix containing 12.5 μL KAPA HiFi HS mastermix (Merck), 0.25 μL 10 μM Smart-seq2 ISPCR primer (SigmaAldrich) and 0.5 μL of a pool of all targeting primers, present at 1 μM each. cDNA amplification was performed by 98 °C 3 min, 21 cycles of [98 °C, 20 sec, 67 °C, 60 sec, 72 °C 6 min], and 72 °C, 5 min. cDNA was the cleaned up using an equal volume (25 μL) of CleanPCR beads (CleanNA) and tagmented using homemade Tn5 (ref. [56]). In brief, cDNA was diluted to ~150 pg/μL, Tn5 was diluted 1:10–1:100 and combined with (20 mM Tris-HCl pH7.5, 20 mM MgCl2, 50% DMF) and diluted DNA in a 1:2:1 volume ratio so total volume of 5 μL. The mix was incubated at 55 °C for 3 min and afterwards shifted to ice and inactivated by the addition of 1.25 μL 0.2% SDS. PCR was performed by adding 6.75 μL of KAPA HiFi HS mastermix, 0.75 μL of DMSO and 1.25 μL each of the forward i5 and reverse i7 library primer (Supplementary Data 4) at 10 μM. PCR program was 72 °C 3 min, 95 °C 30 sec, 12 cycles of [98 °C, 20 sec, 58 °C, 15 sec, 72 °C 30 sec], and 72 °C, 3 min.

**Single-cell cultures**. Bone Marrow mononuclear cells from patient P1 were stained and Lin- or Lin-CD34 + single cells were index-sorted into ultra-low attachment 96-well plates (Corning) containing 100 μL StemSpan SFEM media (Stem Cell Technologies). Media was supplemented with penicillin/streptomycin (100 ng/mL), L-glutamine (100 ng/mL) and the following human cytokines (all from Peprotech): SCF (20 ng/mL), Flt3-L (20 ng/mL), TPO (50 ng/mL), IL-3 (20 ng/mL), IL-6 (20 ng/mL), G-CSF (20 ng/mL), EPO (40 ng/mL), IL-5 (20 ng/mL),

M-CSF (20 ng/mL), GM-CSF (50 ng/mL). After 21 days at 5% CO2 and 37 °C, colonies were imaged by microscopy, and processed as detailed in the following.

**Targeted DNA sequencing by nested PCR amplification**. Single-cell derived colonies were transferred into 50 μL buffer RLT (Qiagen). Cleanup was performed using CleanPCR beads (CleanNA) at a 1.8x volume ratio and eluted in 20 μL 10 mM Tris-HCl pH 7.8. 4.5 μL were transferred to a PCR plate containing 7.5 μL Kapa HiFi HS mastermix and 3 μL of a pool of all outer primers (Supplementary Data 4), each primer at 0.5 μM) were added, followed by a PCR program of 98 °C 3 min, 30 cycles of [98 °C, 20 sec, 63 °C, 60 sec, 72 °C 10 sec] and 72 °C, 5 min and subsequent enzymatic cleanup with 2.5 μL 10x ExoI buffer, 0.4 μL ExoI (NEB) and 0.4 μL FastAP (ThermoFisher), 30 min incubation at 37 °C and 5 min inactivation at 95 °C. Afterwards, 1 μL was transferred to a PCR tube containing 5.9 μL water, 7.5 μL Kapa HiFi HS mastermix and 0.6 μL of a pool of all inner primers (Supplementary Data 4, each primer at 0.5 μM), followed by a PCR program of 98 °C 3 min, 15 cycles of [98 °C, 20 sec, 65 °C, 15 sec, 72 °C 30 sec], 72 °C, 5 min and enzymatic cleanup as above. One microliter was then transferred into a PCR with Nextera indexing primers (Supplementary Data 4) and amplified with 98 °C 3 min, 10 cycles of [98 °C, 20 sec, 60 °C, 15 sec, 72 °C 30 sec] and 72 °C, 5 min.

**Processing of next generation sequencing data**. Raw sequencing reads from MutaSeq and Smart-seq2 experiments were processed using the BBDuk software to trim both the standard Illumina Nextera adapters and the ISPCR adapter. Reads were then mapped to the hg38 human genome (Ensembl release 95) using STAR v2.6 (ref. [57]), with the outFilterMismatchNmax parameter set to 5. Exonic gene counts were tabulated, keeping only reads that did not overlap with targeted regions, overlapped with only one annotated gene, and with lengths greater than 30 nt. For the colony DNA sequencing experiment, reads were mapped to the hg38 human genome using bwa mem (v0.7.17)[49].

**Analysis of mitochondrial mutations and reconstruction of clonal hierarchies (mitoClone package)**. The following set of routines are implemented in the mitoClone package available form https://github.com/veltenlab/mitoClone, and documented further in the package vignettes.

*Construction of allele count tables*. For each cell and position of mitochondrial genome as well as other genomic sites of interest, count tables for all nucleotides (A/C/G/T), and deletions were created from the BAM files. To this end, the mitoClone package implements the baseCountsFromBamList function, which essentially serves as a wrapper to the bam2R function from the deepSNV R package[58].

*Filtering of mitochondrial variants*. To identify relevant somatic variants, we implemented the *mutationCallsFromCohort* function. In short, we select coordinates in the mitochondrial genome containing at least five reads each in at least 20 cells. To distinguish RNA editing events and true mitochondrial mutations at the level of genomic sites, we then identify mitochondrial variants that occur in several individuals. For this purpose only, individual cells are called as "mutant" in a given site of the mitochondrial genome if at least 10% of the reads from that cell were from a minor allele (i.e., distinct from the reference). Mutations present in at least 1% of cells in a given patient, but no more than 10 cells in any other individual, are then included into the final data set and counts supporting the reference and mutant alleles are computed as for sites of interest in the nuclear genome. Mutations present in several individuals are stored as a blacklist and were used further for filtering some of the data analyzed in Supplementary Fig. 4. Importantly, the result from this step is simply a list of genomic sites that are likely to display genetic variability across single cells. The vignette "Variant calling and blacklist creation of the mitoClone package" provides further recommendation for the choice of filtering parameters.

*Construction of clonal hierarchies*. To construct a basic clonal hierarchy, we implemented the muta_cluster function. In short, all nuclear and mitochondrial variants with coverage in at least 20% of cells are selected. Observed variant allele frequencies (VAF) are then computed for each cell. From these values, we compute a ternary matrix of observed variant calls $N_{c,g}$; for each cell $c$ and genomic site $g$, cells were classified as mutant (1) if the VAF was above 5%, reference (0) if it was below 5%, or dropout (NA) if the site was not covered. We thereby assign a genotype "mutant" or "non-mutant" to single cells at all genomic sites selected in the filtering step, using a less stringent cutoff for calling the site as mutant. Then, PhISICS[36] is used to reconstruct a clonal tree. Unlike conventional algorithms for the reconstruction of phylogenetic hierarchies, PhISCS is very robust with regard to noise. For the figures presented in the main text, we ran PhISCS assuming a false-positive rate (FPR) of 3% and an allelic dropout rate (AD) of 10% across all genes. We additionally estimated the dropout rate on a per-gene level from the number of complete dropouts, and varied the resulting parameter vector using Latin hyper-cube sampling around the means using a beta distribution with concentration parameter of 10. Across 80 sampling runs, the same tree was consistently obtained except that the order of nodes within clones was swapped (Supplementary Data 5). PhISCS results are, therefore, robust to variations in the parameters of the

statistical technique used. The false-positive rate of MutaSeq had been empirically estimated using spike-in controls (Supplementary Fig. 10).

*Clustering of mutations into clones and assignment of cells to clones.* PhISCS provides a maximum likelihood phylogenetic tree, enforcing an ordering of mutations. In reality, however, not all intermediate evolutionary steps are represented by cells present in the biological sample (for example, in P1, there are few or no cells displaying the *SRSF2* mutation, but not the mt:11559 G > A mutation). Hence, the order of the nodes in the maximum likelihood tree is to some extent arbitrary and driven by noise; even if there is some statistical support for a specific order, it may be attractive in practice to merge mutations into clones so as to obtain a biologically meaningful, interpretable analysis. We therefore implemented the clusterMetaclones function, which employs a likelihood-based approach. The maximum likelihood tree is split into contiguous linear branches (e.g., for P1, nuc: *SRSF2*, mt:11559G > A, mt:7527DEL and nuc:*EAPP* constitute one such branch). Within each branch, all nodes are then swapped with each other and the likelihood of the data given the altered structure is calculated using the PhISCS model:

$$\mathcal{L} = \prod_c \prod_g \begin{cases} \alpha & \text{if } M_{cg} = 1 \ \& \ N_{cg} = 0 \\ \beta & \text{if } M_{cg} = 0 \ \& \ N_{cg} = 1 \\ 1 & \text{if } N_{cg} = NA \\ (1 - \alpha) * (1 - \beta) & \text{else} \end{cases} \quad (1)$$

Here, $M_{cg}$ indicates whether according to the model, cell $c$ is mutant at genomic site $g$. $\alpha$ is the allelic dropout rate and $\beta$ is the false-positive rate.

The branch is then split into clones such that within each clone, the average difference in log-likelihood incurred by swapping nodes was smaller than 1 per cell. This threshold is set arbitrarily to obtain a practically useful grouping of mutations into clones. The vignette "Computation of clonal hierarchies and clustering of mutations" of the mitoClone package provides further practical recommendations.

The same model was used to compute the likelihood of clonal assignments for each cell. For the analysis in Supplementary Fig. 3b, this estimate was then transformed to bits of information using the Kullback–Leibler distance:

$$D_{KL} = \sum_c \mathcal{L}_c \log_2(\mathcal{L}_c * |C|) \quad (2)$$

where $|C|$ is the total number of clones.

*Quantitative analyses.* For all quantitative analyses of clones (differential expression testing, constribution of clones to cell types) cells with a likelihood of clonal assignment of <0.8 were removed.

*De novo mutation calling.* To identify nuclear mutations associated with the clones, we performed variant calling at a list of candidate sites from COSMIC[37]. We therefore focused on a subset of COSMIC including putatively pathogenic SNVs and small InDels, variants in expressed genes (mean > 20 reads per cell), and variants associated with a primary site "haematopoietic_and_lymphoid_tissue", obtaining a list of 13,797 sites of interest. SNVs commonly observed as germline variants in the 1000 genomes project data set were removed[59]. Allele count tables were created for each site as described above. Finally, a beta-binomial model with the same probability for mutant in all cells was compared to a beta-binomial model with a different probabilities for mutant in each clone using Akaike's Information Criterion.

**Single-cell gene expression data analysis.** Cells with <500 distinct genes observed and genes that appeared in <5 cells were removed. Additionally, data from a healthy individual ("H1") was downloaded from the NCBI Gene Expression Omnibus (GSE75478). Data from all individuals was then integrated using scanorama according to the workflow described by the authors[60]. The low-dimensional data representation obtained by scanorama was then loaded into Seurat[61], and default Seurat implementations of UMAP[62,63] and graph-based clustering were used for data visualization and clustering, respectively, using the first 15 scanorama components.

For more detailed analyses of the T/NK-cell and HSC/MPP populations, cells with these identities were selected and the MNN[64] data integration workflow was repeated using raw expression counts of these specific cell populations as input.

Differential expression testing was performed using MAST[65], using a linear model containing the variable of interest (e.g., clonal identity), a library quality covariate (i.e., the number of genes observed per cell), and, when applicable, additional covariates accounting for patient and cell type.

For the display of gene expression values only, data were normalized according to the Seurat defaults (i.e., divided by the total count of RNA in the cell, multiplied by a scale factor of 10,000 and log-transformation).

**Data visualization.** All plots were generated using the ggplot2 (v. 3.2.1) and pheatmap (v. 1.0.12) packages in R 3.6.2. Boxplots are defined as follows: the middle line corresponds to the median; lower and upper hinges correspond to first and third quartiles. The upper whisker extends from the hinge to the largest value no further than 1.5 * IQR from the hinge (where IQR is the inter-quartile range, or distance between the first and third quartiles). The lower whisker extends from the hinge to the smallest value at most 1.5 * IQR of the hinge. Data beyond the end of the whiskers are called "outlying" points and are plotted individually[66].

**Statistics and reproducibility.** Statistical analyses were performed using R 3.6.2. Statistical details for each experiment are provided in the figure legends. FlowJo v10 TreeStar was used for the analysis of flow cytometry data.

**Reporting summary.** Further information on research design is available in the Nature Research Reporting Summary linked to this article.

## Data availability

Count tables and other processed data necessary to reproduce all analysis from the manuscript are deposited in figshare with https://doi.org/10.6084/m9.figshare.12382685.v1 (ref. [67]). Raw sequencing data are deposited under a Data Access Agreement to protect patient privacy in the European Genome-Phenome Archive with the accession id EGAS00001003414. Requests for data access shall be addressed to LMS (larsms@embl.de). The following restrictions apply: Research with the goal of identifying characteristics of the patient not related to the leukemia (such as surname inference and ancestry research) are excluded. The use of the data for projects not related to cancer research is excluded, exceptions may apply in the context of research aiming to develop new bioinformatics methods. Sequencing data from the healthy individuals are deposited in GEO with the accession id GSE75478. The source data underlying Figs. 1f, 5d, 5h, and Supplementary Fig. 8c–e are provided as a Source Data file. All the other data supporting the findings of this study are available within the article and its supplementary information files and from the corresponding author upon reasonable request. Source data are provided with this paper.

## Code availability

Code for MutaSeq primer design is available at https://github.com/veltenlab/PrimerDesign[68]. A package for data analysis is available at https://github.com/veltenlab/mitoClone[69].

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

## Acknowledgements

We thank Niko Beerenwinkel for discussions, and Andreas Gschwind for contributing code for primer design. We thank the members of the Steinmetz and Haas labs for discussions and support, and we thank the EMBL and DKFZ Genomics core facilities, and the EMBL flow cytometry core facility for technical support. This project was financially supported by the Deutsche José Carreras Leukämie Stiftung grant DJCLS 20R/2017 (to L.V., S.H., L.M.S., and A.T.), the Emerson foundation grant 643577 (to L.V. and L.M.S.) and the German Bundesministerium für Bildung und Forschung (BMBF) through the Juniorverbund in der Systemmedizin "LeukoSyStem" (FKZ 01ZX1911D to L. V., S.H., and S.R). Contributions by S.R. were further supported by Emmy Noether Fellowship RA 3166/1-1 (DFG). Contributions by C.P. were supported by a Max-Eder Grant (German Cancer Aid 70111531). Contributions by D.N., J.C.J., W.K.H., and T.B. were supported by the Gutermuth Foundation, the H.W. & J. Hector fund, Baden-Württemberg, and the Dr. Rolf M. Schwiete Fund, Mannheim. D.N. is an endowed professor of the Deutsche José Carreras Leukämie Stiftung (DJCLS H 03/01).

## Author contributions

L.V. developed MutaSeq and performed single-cell RNA-seq experiments with assistance by J.M., D.R.L., and C.S.T. P.H.M. performed sample characterizations, single-cell culture experiments, and FACS sorting with support by M.P., A.D., S.R., and supervision by S.H. B.A.S. and L.V. developed mitoClone and analyzed the data with contributions from R.F. L.V., S.H., A.T., and L.M.S. conceived the study. L.V. and B.A.S. wrote the manuscript with contributions from P.H.M., S.H., S.R., and L.M.S. C.L., D.N., J.C.J., C.P., T.B., W.K. H., and C.M.T. collected samples and performed initial sample characterization. All authors have read and commented on the manuscript.

## Competing interests

L.M.S. is co-founder of Sophia Genetics and Levitas Bio and consultant for several companies on genetic analysis. All other authors declare no competing interests.
