## [Peer Review File · Nature Communications]

REVIEWER COMMENTS

Reviewer #1 (Remarks to the Author):

In this manuscript, Velten and colleagues describe a single cell genomic approach that integrates transcriptomics with clonal tracking. By applying their methodology to samples from four patients with acute myeloid leukemia, the authors suggest that their combined nuclear and mitochondrial genotyping approach enables the de novo identification of pre-leukemic, leukemic, and non-leukemic clones within the transcriptionally resolved hematopoietic hierarchy. The authors further analyze gene expression characteristics of these cells and relate these to healthy progenitor populations. The integration of clonal information into single cell genomic readouts is an important problem and the study as presented is technically sound. However, the authors weaken the manuscript by making inaccurate claims, particularly about what is truly novel in this work. Many similar findings have been previously reported by other groups and the value of the paper is reduced by misleading statements. Moreover, the authors focus on a heterogeneous group of four patients with acute myeloid leukemia and the biological insights that can be gained through such a limited set of samples are unclear. Some of the biological implications seem overstated, given such a small sample set.

Major Comments:

1) Several statements in the manuscript are misleading and should be revised to more accurately reflect the capacities and limitations of the methodology. For example:

"MutaSeq does not rely on previously known nuclear mutations to detect clones";

"[...] pre-leukemic, leukemic and non-leukemic clones were identified de novo and distinguished unanimously";

"However, these protocols suffer from excessive dropouts, are of low throughput, and/or require prior knowledge of genomic mutation sites."

Generally, single cell RNA-seq approaches inherently suffer from false positive mutations (e.g. high error rates of endogenous RNA polymerases during reverse transcription) that have to be accounted for. The authors themselves integrate information from exome sequencing results to ensure reliable mutation calling. In fact, this appears essential as the detection of nuclear mutations is utilized to a priori define clones as pre-leukemic or leukemic. Furthermore, the detection of a specific mutation requires the design of primers to flank the cDNA region that encompass the mutations of interest. Therefore, it is unclear how clones can be detected without previously knowing the nature of specific nuclear mutations, other than potentially relying on mitochondrial mutations alone. In the presented context of acute myeloid leukemia however, it appears quite essential to be able to detect nuclear mutations.

Moreover, as the authors show, not every mutation may be reliably detected (e.g. when the gene is lowly expressed or with frameshift mutations) and there may be instances, where the methodology may not be able to allow unanimous delineation of clonal groups. In this regard statements like "de novo" and "unanimously" do not appear justified, and it is unclear how Muta-seq in fact overcomes the limitations of other approaches. Dropouts, the limited throughput of plate-based approaches such as Smart-seq2 (compared to Reference 26), and the need for a priori knowledge of nuclear genomic mutations still present limitations that also apply to Muta-seq. Therefore, the added value of this approach is unclear and many claims seem overstated.

2) The authors should discuss how their approach is distinct or superior compared to TARGET-seq (Reference 29), which is conceptually similar, is also based on Smart-Seq2, and shows high mutation detection efficiency. Potentially Muta-seq has higher multiplexing capabilities to detect a larger number of nuclear mutations, but this has not been fully taken advantage of in the work as presented. While the integration of somatic mitochondrial mutations adds depth, in theory their detection would also be possible using the TARGET-seq approach (as this is Smart-Seq2 based), the mitochondrial reads may simply not have been considered at the time. Moreover, integration of T cell receptor sequences and BCR-ABL fusion transcripts from cDNA have previously been

accomplished and integrated with mitochondrial mutation calling with Smart-Seq2 (Reference 31). The computational approach may be more unique here, though it is unclear how it compares to what has previously been attempted (please see the next comment, as well).

3) The authors state:

"Here we take advantage of the observation that bonafide mitochondrial mutations are exclusive to individual patients, whereas RNA editing and other base modifications are shared across individuals. Only those mitochondrial mutations uniquely occurring in individual patients are used for downstream clonal tracking."

This is an interesting approach and while the reasoning appears intuitive, it appears to be primarily benchmarked against a single exome sequencing dataset (Figure S2e,f). It is also unclear whether this approach would make orthogonal DNA-based filtering obsolete or whether it would still be recommended. Notably, allele frequencies appear to deviate quite substantially in the scRNA vs. exome sequencing data (Figure S2e), e.g. variants with an allele frequency of >0.3 in scRNA are only present at allele frequencies of <0.05 in the exome data. This appears to be in contrast to allele frequencies of nuclear mutations that appear more concordant (Figure 1f). How do the authors explain this discrepancy? Conceivably the exome capture is not designed to also retrieve mitochondrial genomic information? How does their computational approach compare in additional datasets, e.g. Smart-Seq2 data presented in Figures 5 and 6 from Reference 31? Could the mitochondrial variants enhance findings, add resolution to data from Reference 29?

4) In Figures 4 and 5 the authors provide evidence about how clonal tracking information can enhance single cell transcriptomics data. The authors state:

"Together, these results demonstrate that clonal tracking of single-cell transcriptomes can identify the molecular effects of clonal evolution in human, orthogonal to the use of mouse models^{34,37-39} but with consistent results."

The consistency of results obtained from a limited group of heterogeneous patients is unclear. To truly make such a strong statement, the authors would need to examine many additional acute myeloid leukemia samples.

5) Fig. 2e/f displaying the results of the PhISCS approach appear to imply an order on how the various mutations were acquired. Many mitochondrial mutations however appear to be shared within a single clone and are otherwise not subclonal. It is not intuitive to follow how a temporal order may therefore be readily delineated here?

6) Some parts of the introduction and discussion make inaccurate claims that should be corrected. For instance, the authors state, "However, the latter studies have focused on analysis of single-cell ATAC-seq data..." However, the value of mitochondrial mutation calling in Smart-Seq2 has already been shown in Reference 31.

Reviewer #2 (Remarks to the Author):

In this work, Velten et al describe a modified version of the SmartSeq2 protocol, MutaSeq, that is able to define clones and associate targeted mutagenic information without affecting the overall performance of the original protocol. They analyse four AML patients using this methodology to define leukaemic, pre-leukaemic and healthy clones and use the information to get a better understanding of the biology of these clones.

General Comments

The authors carefully looked into the implications of introducing additional oligonucleotides in the amplification reaction for the overall results of the SmartSeq2 protocol and followed an interesting approach by associating genomic mutations to mitochondrial mutations. However, there are

certainly important limitations and concerns:

- 1) MutaSeq improves the dropout rate of SmartSeq2 but the dropout rate is still very high for a large number of regions (as depicted in figure 1e). The detection rate will strongly depend not only of the expression levels of the gene (figure 1d), but also of the distance of the mutation to the polyadenylation site and the complexity of the sequence of the gene of interest. The high dropout rate makes the association of the genomic and mitochondrial mutations difficult.
- 2) The co-detection of genomic mutations is low (4-5 mutations out of 14 per cell according to figures 1c and S1a). The authors do not show if this is due to technical performance of the primer pool or due to coverage.
- 3) Previous knowledge is required for the detection of the specific genomic mutations to be analysed. However, it is worth mentioning that MutaSeq would allow retrospective analysis of additional mutations.
- 4) The method may be difficult to be upscaled to droplet methods, which are strongly biased towards either 3' or 5' regions, relying in more expensive technologies such as SmartSeq2.
- 5) It requires relative deep sequencing (~788,000 reads per cell).
- 6) The authors assume that cancer stem cells cannot be separated from healthy stem cells based solely on gene expression using a healthy donor as a reference. The authors should investigate if a focused analysis of the HSPC compartment would be able to separate the cancer stem cells from healthy stem cells in the different patients without relying on mutational information and then compare with the mutational information.

The authors state that the method do not rely on previously known nuclear mutations to detect clones. However, although the statement is technically true, they could not define clones using the mitochondrial mutations in 2 of the patients (50% of the analysed samples!!). Additionally, the association to the genomic mutations is not great because of the dropouts (see below).

Overall in the manuscript, the Figure legends are quite bare and not very descriptive which makes the interpretation difficult and more laborious. A much more detailed description of the figures is needed in the figure legends, clearly describing what is shown in each panel and how the data was obtained. Definition of the scales in the plots is also typically absent. The output for all the comparisons performed in this work should also be included (e.g. comparison of CD34- blasts to HSC/MPP-like cells), all data relevant to the article should be deposited (including DNA exome results for colony assays in P1) and the code must be made available.

Specific Comments

- 1) The authors indicate in the introduction section that "previous protocols suffer from excessive dropouts, are of low throughput and /or require prior knowledge of genomic mutation sites". This is also the case for this method as well.
- 2) S1b should be plotted in a normalised way, to be able to compare how many reads are on target (for example in reads/100,000 of mapped reads) per cell.
- 3) Figure legends of figures S1c and S1d should explicitly indicate the meaning of the concentrations.
- 4) Figure 1e.- Define "reference" vs "mutant". Why does the proportion of the call of the mutation change drastically according to the method used in some genes, such as SRSF2? How confident are the authors that the "reference" or "mutant" call is not to affected by dropouts of the other allele?
- 5) Why is it that MutaSeq and SmartSeq2 are drastically over estimating some of the point mutations in Figure 1f?
- 6) The authors state that both methods underestimate the abundance of frameshift mutations, likely due to nonsense mediated decay. Do the authors actually have evidence for this claim?
- 7) The authors should tone done the statement "MutaSeq efficiently covers both genomic target sites and the mitochondrial genome in single-cell RNA-sequencing experiments". The method improves the results when compared to SmartSeq2 but it has a high dropout rate in a large number of genes.
- 8) The authors define preleukaemic and leukaemic mutations but they should better explain/demonstrate/reference the rationale behind the assignment for each mutation. Also, the first description of the status affiliation of the mutations appears at the very end of heading

"Simultaneous mapping..." but there are references to this classification and separation much earlier in the text. Please clarify earlier on to gain in clarity of the article. The authors also mention that leukaemic cells often exist in a cell state that is similar to healthy HSC/MPPs. Could it be because these are cells that contain the mutation but they are not transformed into leukaemic cells?

9) The detection of the DNMT3A mutation present multitude of dropouts and the association values to the mitochondrial mutations are very low. Therefore it is difficult to make any statements or claims about the preleukaemic status of cells in patient P2.

10) It is unclear in the main text which cells were analysed from which patient. It would be useful to have in the main text a brief description of the cells sorted and the quantity of each type used for the analysis.

11) Include percentages in the boxes in Figure S3a.

12) Why weren't all populations in Figure S3a analysed for the 4 patients? E.g. Cd34-Lin+ cells were not analysed for P1 or P4? Why several of the gates are different between the healthy control and patients, such as FSC vs SSC (healthy) or CD34+ gate for patient 3. Can the authors explain the rationale?

13) An additional UMAP plot should be added to the supplementary figures where cells are coloured according to the gating strategy for each patient.

14) Please indicate the methodology followed to obtain the clusters in figures 2b, 2c and 2d.

15) Could the authors show an example of how the data looked without the mitochondrial mutations included (Fig 2b and d)?

16) The authors claim that there are mutually exclusive mutations such as KLF7 and CEBPA. However, it is difficult to define the mutation status of KLF7 in P1 and P3 (or TET2 and even SPEN in other patients) due to the high rate of dropouts (as expected from data in Figure 1). How certain can the authors be of this claim?

17) The reference or demonstration of the fact that only bona fide mitochondrial mutations are specific to individuals should be shown.

18) The description of the analysis of mitochondrial mutations is not completely clear. In particular, I find confusing the sentence "Individual cells were then called as mutant in these sites if at least 10% of the reads from that cell supported a minor allele.". Please clarify.

19) The information relative to the genomic mutation status for TET2 and SPEN is not used in Figure 2 for P1. Was this due to the very high rate of dropouts?

20) The "healthy" clone of patient P1 is clearly negative for preleukaemic mutation SRF2 but the cells for this clone also have the common characteristic of being mostly dropouts for Cebpa. It would be interesting to increase the number of SRF2-negative colonies in the DNA-seq (Figure 2d) to fully verify that they are CEBPA-negative.

21) The legend of figure S2e is not complete.

22) Key legends in figure S2f are swapped.

23) Add the meaning of grey dots in key legend of figure S2g.

24) What cell type is the pale yellow cluster in Figure 3a?

25) The population within Fig 3a labelled as "unclear" would be better labelled as "unknown" or "unclassified".

26) It is difficult to distinguish HSC/MPP population from others in the healthy donor in Fig 3c. Please use a different colour.

27) It is unclear the cells that were used to perform the comparison in Figure 3g. Are the cells within the pale yellow cluster included in the category "CD34+ blasts" or only the cells within the dark yellow cluster? Please specify and justify the selection. In a similar way, can the authors specify which cells were used for the comparisons in figure 3h? According to Table S3, 162 genes obtained a FDR < 0.1; however, the Venn diagram in Figure 3h shows 639 genes. How is that possible? According to the Venn diagram in Figure 3h, there are much fewer upregulated genes for Cd34- blasts. Is this population then more similar to the HSC/MPP? What about the downregulated genes?

28) Where is the unsupervised analysis to separate HLF+, GATA2+, Flt3+ cells? Please show the output of the clustering. Also include RNA expression of genes HLF and Flt3 (in addition to surface marker) in fig S4a.

29) Gene name S1008 should be included in text as gene for calprotectin.

30) It is very difficult to appreciate the differences in expression levels in Figure S4a. A different colour code should be used.

31) Would it not be better to show a specific marker which highlights the identification of each

lineage in FigS4a? Along the same lines, Tal1 can be removed as very few cells seem to express this transcription factor.

32) The statement "Furthermore, all patients retain cells highly similar to healthy HSCs." should be modified to reflect that it is to a different degree in the different patients.

33) Please confirm what the dashed line (eclipse) is highlighting in figures 4a to 4d. It is shifted in patients P3 and P4.

34) The authors do not show sufficient data about their capacity to detect IDH2 nor NUP188, in terms of dropout proportion. The data in this respect in figures 4c and 4d is very difficult to appreciate. Please include similar data to figure 1e.

35) In Figure 4c, it appears as if the higher allele frequency for the leukemic cells (as IDH2) is in the lymphoid cluster (T cells/NK cells) but figure 4e indicates that 50% of the cells in the T-cell cluster are non-mutated. How is this possible?

36) The authors state that "Across patients, leukemic cells were not observed in lymphoid (B, NK, T) lineages" yet figure 4e clearly shows the presence of leukemic cells in the T cluster of cells patients P1, P2 and P3.

37) How significant are the claims that CD96 is expressed higher in leukemic P3, when there are only approximately 8 cells within this category?

38) The authors need to take care, they describe the SRSF2 mutation as pre-leukemic but then subsequently state that "In patients P1 and P3 (SRSF2 mutated), leukemic cells". I believe that the authors mean that P1 and P3 have the SRSF2 mutation, but it is misleading as currently stated.

39) The authors claim in P4 the ability to contribute to the erythroid lineage is restricted to pre-leukemic and non-leukemic clones, however figure 4e show the presence of leukaemic cells in the MEP/erythroid cluster.

40) The authors claim that in P4 mutations were only observed in CD34- blasts (and so is reflected in figure 4e) however, the UMAP in Fig 4d shows that the allele frequency for NUP188 is 1.0, with high log₁₀ read counts in the HSC/MPP region of the UMAP. Can the authors please explain this and clarify in the text.

41) In Figure 4e it is important to add information about the total number of cells in each category for each patient.

42) Also in Figure 4e, I wonder if the category "Blasts (CD34- HBZ+)" is labelled correctly since according to figure S4a the expression of HBZ is very restricted outside some CD34+ blasts.

43) Figure 4e shows that all cells in HSC/MPP compartment of patient P4 are "non AML". However, figure 4d shows allele frequency close to 1 for NUP188 for most cells in the HSC/MPP compartment. How can this be possible?

44) The authors should tone down the statement "Together, these results demonstrate that leukaemic mutations may initiate differentiation blocks at various levels." Although the results suggest that this could be the case, no additional experiments to demonstrate such blockage in differentiation were performed.

45) In figure 5a, the text indicates "high" expression of HFL but data shows barely below 1. Please define the scale of expression.

46) In Figure 5a there seems to be a typo in the legend, since it seems to relate to patient P2 and not to patient P3.

47) In Figure 5a, indicate number of cells for each category. It looks like there are very few cells which could affect the interpretation.

48) How many cells are involved in the comparisons showed in figure 5b? Are enough cells to reach enough statistical power to make those comparisons? This is a general concern across all volcano plots shown in the article and should be addressed.

49) It is difficult to appreciate which genes are differentially expressed in figure 5b. Because of the scale, it looks like Hoxb3/CD96 have a logFC close to 0. Indicate the thresholds in the figure and include full results as supplementary data.

50) The authors discuss the role of KLF7 mutation in HSPC cells from patient P1 (Fig 5d). How many cells are included in the each category of the comparison? Have they also looked at the CEBPa clone, as these cells also occupy the same cellular population/state?

51) Include full results for the comparison of pre- and leukaemic cells. Indicate the number of cells in each category. Also, was it done by patient and/or merging patients?

52) It would be interesting to know how many cells were included in each category of the comparison of (pre-)leukemic vs non-AML cells. It is peculiar that so few genes came up as differentially expressed for P2 taking into account that these cells appear in very different locations

of the UMAP. Also, it is curious that only FOS is upregulated out of the rest of AP1 factors.

53) Please include in Table 1 the data regarding to which regions were targeted for patients P2 and P4. Describe the nature of highlighted genes in this table.

54) Consistency is required throughout, the authors use pre-leukemic and preleukemic, one should be chosen and used throughout the manuscript.

55) After multiple occurrences of a colon in the text a capital letter is used, this should be corrected.

56) The authors should consider to re-order the panels within the figures and in some cases the figures themselves to ensure that they are referred to in a chronological order throughout the text (for example figure S2c is mentioned after S2g, fig S3 is mentioned before S2).

57) Maybe "Unanimously" should be swapped to "unequivocally".

REVIEWER COMMENTS

Reviewer #1 (Remarks to the Author):

In this manuscript, Velten and colleagues describe a single cell genomic approach that integrates transcriptomics with clonal tracking. By applying their methodology to samples from four patients with acute myeloid leukemia, the authors suggest that their combined nuclear and mitochondrial genotyping approach enables the de novo identification of pre-leukemic, leukemic, and non-leukemic clones within the transcriptionally resolved hematopoietic hierarchy. The authors further analyze gene expression characteristics of these cells and relate these to healthy progenitor populations. The integration of clonal information into single cell genomic readouts is an important problem and the study as presented is technically sound. However, the authors weaken the manuscript by making inaccurate claims, particularly about what is truly novel in this work. Many similar findings have been previously reported by other groups and the value of the paper is reduced by misleading statements. Moreover, the authors focus on a heterogeneous group of four patients with acute myeloid leukemia and the biological insights that can be gained through such a limited set of samples are unclear. Some of the biological implications seem overstated, given such a small sample set.

We thank the reviewer for this summary of our work. To avoid misleading statements on novelty, we have carefully gone over the text and now more clearly define the precise advances made by our study, as compared to previous publications (in particular, Ludwig et al., Cell 2019 for the identification of clones from single-cell RNA or ATAC sequencing data, and Rodriguez-Meira, Molecular Cell 2019 for nuclear mutation calling). To summarize key changes made in the revised version (see below for a detailed response to each of the reviewer's points):

- While our approach contains technical advances over existing methods, key novelty lies in the adaptation of these methods to identify and characterize cancer stem cells, which we now make clearer in our revised abstract, introduction and discussion.
- Regarding technical novelty in the context of clonal tracking, as suggested by the reviewer we have further investigated the ability of our approach to **identify and characterize clones de novo in an unsupervised manner** (New Supp. Figures 4,6, and a new subchapter in the results section).
- We believe we are the first to provide the required computational routines for the *de novo* identification and characterization of clones from single-cell RNA-seq data. We now provide an R package *mitoClone* for the community that implements our entire computational workflow and we demonstrate that this package can be used to identify and characterize clones in Smart-seq2 or MutaSeq data of single patients, with no need for a reference (New Supp. Figure 4).
- We have expanded the discussion of the technical advances made by our study, as well as its limitations (summarized in new Supp. Figure 9).
- Finally, we have made it clear in the discussion and results part of the manuscript that our analysis of a small patient population represents a demonstration of MutaSeq's power, but that studies in a larger cohort are required to assess their general validity and relevance to AML.

Major Comments:

1) Several statements in the manuscript are misleading and should be revised to more accurately reflect the capacities and limitations of the methodology. For example:

"MutaSeq does not rely on previously known nuclear mutations to detect clones";

"[...] pre-leukemic, leukemic and non-leukemic clones were identified de novo and distinguished unanimously";

"However, these protocols suffer from excessive dropouts, are of low throughput, and/or require prior knowledge of genomic mutation sites."

Generally, single cell RNA-seq approaches inherently suffer from false positive mutations (e.g. high error rates of endogenous RNA polymerases during reverse transcription) that have to be accounted for. The authors themselves integrate information from exome sequencing results to ensure reliable mutation calling. In fact, this appears essential as the detection of nuclear mutations is utilized to a priori define clones as pre-leukemic or leukemic. Furthermore, the detection of a specific mutation requires the design of primers to flank the cDNA region that encompass the mutations of interest. Therefore, it is unclear how clones can be detected without previously knowing the nature of specific nuclear mutations, other than potentially relying on mitochondrial mutations alone. In the presented context of acute myeloid leukemia however, it appears quite essential to be able to detect nuclear mutations.

Moreover, as the authors show, not every mutation may be reliably detected (e.g. when the gene is lowly expressed or with frameshift mutations) and there may be instances, where the methodology may not be able to allow unanimous delineation of clonal groups. In this regard statements like "de novo" and "unanimously" do not appear justified, and it is unclear how Muta-seq in fact overcomes the limitations of other approaches. Dropouts,

the limited throughput of plate-based approaches such as Smart-seq2 (compared to Reference 26), and the need for a priori knowledge of nuclear genomic mutations still present limitations that also apply to Muta-seq. Therefore, the added value of this approach is unclear and many claims seem overstated.

We appreciate these important points, and have removed misleading statements from the revised manuscript and more carefully explained the capacity and limitations throughout.

Importantly, we have added further evidence to support the point that, albeit only in the presence of mitochondrial somatic variability, our approach allows for a de novo identification and characterization of clones with no reliance on prior knowledge of nuclear mutations. We have included a new sub-chapter and a new supplementary figure S6 to the results (lines 200-224):

We next investigated if the use of mitochondrial somatic variants enables the identification and characterization of clones without prior knowledge of nuclear mutations. To that end, we made use of a dataset from patient P1 generated without amplification of nuclear sites (i.e. standard Smart-seq2). A clear clonal structure was identified in an unsupervised manner based solely on mitochondrial variants (Figure S6a). In order to examine whether the presence of somatic genetic variability is associated with the different clones, we then queried the mutational status of 13,797 genomic sites annotated as mutated in AML in the COSMIC database³⁷ using a beta-binomial model (see Methods). This unsupervised analysis revealed a highly significant association of the SRSF2 P95H mutation with the leukemic and pre-leukemic sub-clones (Figure S6b,c). Their malignant nature was further evidenced by a markedly reduced ability to contribute to the T cell lineage (Figure S6d and see also below).

To further demonstrate our ability to identify clones de novo, we highlight the identification of an expanded CD3+ T-cell clone in P1, and a clonal expansion of non-leukemic cells in P2 (Figure 2e,f). These clones would have been missed by approaches relying on genomic mutations alone^{27,28,31}. The latter event in particular is of interest, since these cells were not associated with the pre-leukemic DNMT3A mutation. By again querying sites from the COSMIC database³⁷ using a beta-binomial model, we identified that they had uniquely acquired a mutation in the RPL3 gene (Figure S6b,e). These results suggest that this clonal expansion event is independent of the leukemia and associated with the acquisition of unrelated nuclear mutations.

Taken together, these results demonstrate that our approach can identify and characterize clones de novo without prior knowledge of nuclear genomic mutations. The mitoClone package implements all routines for clonal clustering and mutation calling.

Response Figure 1: De novo calling and characterization of clones. Included in the manuscript as Figure S6. For an implementation and for reproducing the computations, see the package vignettes of the mitoClone package. a. Unsupervised clustering of mitochondrial mutations identified from a Smart-seq2 dataset of n=672 cells from patient P1. b. De novo identification of nuclear somatic variants associated with the clonal labels from panel a. Difference in Aikake's Information Criterion (AIC) is shown for a comparison between a model where allele frequencies are the same across all cells, and a model where allele frequencies differ between clones. Red line highlights the intercept. See Methods section Analysis of mitochondrial mutations and reconstruction of clonal

hierarchies. c. Boxplot of single-cell allele frequencies for the SRSF2 P95H mutation summarized between clones (Smart-seq2 data). d. Bar plot of contribution of clones to T cells identified from unsupervised clustering of gene expression data (see also Figure 3). e. Boxplot of single-cell allele frequencies for the RPL3 mutation (COSV53365368) summarized between clones (MutaSeq data). Red dashed line highlights the allele frequency of the mutation identified in exome sequencing.

Whether a statistically confident distinction between clones is possible depends on coverage, but more crucially on the presence of mitochondrial somatic variability. In all cases where somatic variability was identified (P1, P2, and additional datasets from Ludwig et al., see also below), the identification of clones was statistically unanimous, and the majority of cells could be assigned to clones with high statistical confidence (see Figure 2 and also S3b for a more focused analysis). As mentioned by the reviewer, in other instances (P3, P4), no unanimous assignment was possible. Accordingly, we now clearly state in the manuscript that in the absence of mitochondrial variability, only qualitative statements can be drawn, comparable e.g. to what is possible with GoT-seq (Nam et al., ref 28).

Line 403-406: In the absence of mitochondrial somatic variation, the MutaSeq protocol can only be used to draw qualitative statements on clonal differentiation capacities (similar to ref. 28, and with an improvement over smart-seq2), but due to dropout, neither enables the statistically confident assignment of cells to clones, nor differential expression testing between clones.

We now also present this limitation more explicitly in the introduction and results sections (lines 93, 198). We also consistently use the term 'statistically clear' (instead of 'unanimous') only when referring to cells assigned to clones using mitochondrial information:

*Line 91-93: Here, we introduce MutaSeq, a workflow that amplifies nuclear mutations from cDNA, and mitoClone, a computational tool that **achieves high-confidence clonal assignments and de novo discovery of clones using mitochondrial marker mutations when available.***

*Line 196-198: Taken together, our approach allows for the identification of putatively leukemic, pre-leukemic, and healthy clones and can assign cells to clones with high confidence **if mitochondrial somatic variation is present.***

*Line 347-348: **In the presence of mitochondrial somatic variability,** our approach does not rely on previously known nuclear mutations to detect clones.*

Line 402-403: The major limitation of our pipeline is that it requires natural somatic variability to resolve clones at high confidence.

And we have reformulated the statement on previous methods in the introduction:

Line 86-89: However, the application of these methods to characterize leukemic stem cells has not been demonstrated, and in particular requires an ability to reliably detect clonal expansion events, to associate clinically-relevant coding mutations to clones with high confidence, and to draw statements on changes in gene expression between clones.

2) The authors should discuss how their approach is distinct or superior compared to TARGET-seq (Reference 29), which is conceptually similar, is also based on Smart-Seq2, and shows high mutation detection efficiency. Potentially Muta-seq has higher multiplexing capabilities to detect a larger number of nuclear mutations, but this has not been fully taken advantage of in the work as presented. While the integration of somatic mitochondrial mutations adds depth, in theory their detection would also be possible using the TARGET-seq approach (as this is Smart-Seq2 based), the mitochondrial reads may simply not have been considered at the time. Moreover, integration of T cell receptor sequences and BCR-ABL fusion transcripts from cDNA have previously been accomplished and integrated with mitochondrial mutation calling with Smart-Seq2 (Reference 31). The computational approach may be more unique here, though it is unclear how it compares to what has previously been attempted (please see the next comment, as well).

TARGET-seq exists in two variants, full-length TARGET-seq and 3' TARGET-seq. For full-length TARGET-seq the detection of mitochondrial variability is certainly possible in theory, however the published data contains an insufficient numbers of cells (<96) reflecting the low throughput of this protocol. 3' TARGET-seq is the main focus of reference 29 and lacks coverage of mitochondria due to its 3' bias (see modified figure S1j, also shown below).

Response Figure 2. Logarithmic coverage of the mitochondrial genome compared between different methods^{27,31,66}. For the plot on the right, coverage was normalized to the number of reads aligning to the transcriptome. Included in the manuscript as figure S1j.

Therefore, these existing datasets cannot exploit mitochondrial mutations to annotate or discover clones. For example, a de novo discovery and characterization of clones in the absence of prior knowledge, such as presented in Figure S6, would not have been possible. We now describe the distinctions of our approach from TARGET-seq and other methods in the discussion and provide a summary in the new supplementary figure S9:

Lines 392-400: Finally, while TARGET-seq addresses the limitation of dropout of nuclear sites, depending on the implementation it does not offer sufficient mitochondrial coverage, or is of very limited throughput. Of all methods available to date, the approach presented here converges on crucial aspects, specifically: a) high mitochondrial coverage, allowing us to identify benign expanded clones de novo, even in the absence of known genetic markers, b) decreased dropout of relevant genomic mutations, permitting the association of clones with genomic mutations, if present and c) a highly confident assignment of cells to clones, enabling quantitative analyses of clone-specific gene expression (Figure S9). These capabilities expand the potential applications of our approach to the study of clonal dynamics during ageing and oncogenesis beyond the hematopoietic system.

	with mitoClone package				no mitoClone package		
	MutaSeq	SmartSeq	TARGET-seq Full L.	TARGET-seq 3'	GoT-seq	SmartSeq	ATAC-seq
Identification of clones (de novo)	✓✓*	✓✓*	0*	✗	✗	✗	✓✓*
Association of nuclear and mitochondrial variants	✓✓*	0*	0*	✗	✗	✗	✗
Identification of clones (known variants)	✓	0	✓✓	✓✓	✓	0	✗
Differential expression testing between clones	✓✓*	✓✓*	✓✓	✓✓	✗	✗	✗
Throughput	✓	✓	✗	✓	✓✓	✓	✓✓

* If natural mitochondrial somatic variability is present.

Response Figure 3: Overview of capabilities of MutaSeq compared to related methods. Included in the manuscript as Figure S9. SmartSeq: ref. 33,53. TARGET-seq: ref. 31. GoT-seq: ref. 28. "0" means theoretically possible, but unproven and/or very limited.

Regarding the suggestion that potentially, Muta-seq might have higher multiplexing capabilities to detect a larger number of nuclear mutations, we have benchmarked multiplexing capacity in Figure S1e and found an optimal performance of MutaSeq when including up to 20-30 primers. However, a similar analysis is not available for TARGET-seq, making any comparison difficult.

Regarding the comments on Reference 33 (formerly ref. 31, Ludwig et al.), please see the next point.

3) The authors state:

“Here we take advantage of the observation that bonafide mitochondrial mutations are exclusive to individual patients, whereas RNA editing and other base modifications are shared across individuals. Only those mitochondrial mutations uniquely occurring in individual patients are used for downstream clonal tracking.”

This is an interesting approach and while the reasoning appears intuitive, it appears to be primarily benchmarked against a single exome sequencing dataset (Figure S2e,f). It is also unclear whether this approach would make orthogonal DNA-based filtering obsolete or whether it would still be recommended. Notably, allele frequencies appear to deviate quite substantially in the scRNA vs. exome sequencing data (Figure S2e), e.g. variants with an allele frequency of >0.3 in scRNA are only present at allele frequencies of <0.05 in the exome data. This appears to be in contrast to allele frequencies of nuclear mutations that appear more concordant (Figure 1f). How do the authors explain this discrepancy? Conceivably the exome capture is not designed to also retrieve mitochondrial genomic information? How does their computational approach compare in additional datasets, e.g. Smart-Seq2 data presented in Figures 5 and 6 from Reference 31? Could the mitochondrial variants enhance findings, add resolution to data from Reference 29?

We thank the reviewer for this suggestion. We have now evaluated our computational approach on publicly available datasets from reference 33 (formerly ref. 31, Ludwig et al.). Reference 31 (formerly ref. 29, TARGET-seq) is discussed in response to the reviewer’s previous point.

These analyses are included in a new figure S4 (see also below) and described in the results as follows:

Lines 168-174: *We further analyzed various control datasets with known associations between mitochondrial mutations and clones³³ to validate that this approach enables the detection of relevant mitochondrial mutations without a need for a DNA-based reference, and further enables the unsupervised identification of clones (Figure S4d-f). We have implemented all the required filtering and blacklisting routines required for the identification of high-confidence somatic mitochondrial variants in the mitoClone R package (<https://github.com/veltenlab/mitoClone>).*

The figure legend of figure S4 provides more detail:

a. Overview of the computational strategy used. In the case of data from a group of individuals, sites were filtered based on coverage and annotated as ‘mutant’ if a specified fraction of reads deviates from the reference allele. Sites were then excluded as likely RNA editing events if the same mutation was observed in more than one individual. Alternatively, in the case of data from a single individual, similar coverage based filtering were applied and data was then filtered against a blacklist created from a cohort. d. De novo variant calling and clustering of a CML patient dataset. Data from ref. 66 were processed and clustered with the mitoClone package. The same variant filtering approach used on the patients from our study was used. Thereby, two patients with substantial mitochondrial variability were identified and in both cases clones associated with the BCR-ABL mutation were resolved in an unsupervised manner. The analysis by ref. 33 had missed one of these patients, did not achieve an unsupervised separation of BCR-ABL+ and BCR-ABL- cells in either case (Figure 7G in ref. 33), and instead relied on stratifying cells by the existing BCR-ABL label (Figure 7J in ref. 33).

To extend the utility of our approach, we additionally sought to modify it such that it would also work on datasets from single patients (and not groups of patients). We therefore compiled a blacklist of false positive variants (i.e. variants shared between individuals) from our patient cohort, and attempted filtering datasets from single individuals based on this blacklist. Again, these results are documented in figure S4:

e. De novo variant calling and clustering of single cells from hematopoietic colonies derived from a single individual. Data from Figure 5 of ref. 33 were processed and clustered with the mitoClone package. Left panel shows unsupervised clustering of mutations identified by the mitoClone package, right panel quantitatively compares unsupervised clustering and colony labels. 27 of the colonies were identified in an unsupervised manner. The analysis by ref. 33 had identified approximately half that number by unsupervised analyses (their Figure 5E), and using supervised methods identified mitochondrial mutations associated with 33 clones (their Figure 5H).

f. De novo variant calling and clustering of single cells from a single colorectal cancer patient. Data from Figure 7 of ref. 33 were processed and clustered with the mitoClone package. Clustering structure obtained by PhISCS is shown and compared to the clustering presented in ref. 33 (row labeled ‘original’), which was based on variant filtering using a DNA-seq based reference. Despite the different filtering approaches, our unsupervised clustering separated the clusters identified by Ludwig et al. and identified additional variability.

Data for Ludwig et al., Figure 6 E-G are only accessible through a data access agreement with Beijing university, who did not respond to our data access request.

The poor correlation between scRNA-seq based and exome-based allele frequencies (Figure S4b,c, formerly figure S2e,f) was expected, since the sequencing was performed on different starting

populations (exome: total bone marrow, RNA-seq: enrichment for CD34+ cells). However, there is qualitative overlap in the mutations the methods were able to identify. This information was added to the figure legend:

b. Allele frequencies and coverage of mitochondrial mutations from P2 in single-cell RNA-seq data compared to whole exome sequencing data (WES). Sites with less than 10 reads per cell in RNA-seq were classified as 'not expressed'. Variants that were observed in WES were classified as 'validated' (circles), other variants were classified as 'not validated' (triangles). The low correlation between the two datasets is likely due to different starting cell populations (WES: Total bone marrow, single cell RNA-seq: enriched for CD34+ cells), and data are only used for qualitative statements (presence/absence of mutations).

c. Bar charts summarizing the classifications from panel b. Left, mutation sites are split by their label based on the mitoClone pipeline; right, sites are split by whether they were detected in WES.

These results, together with the validation of the clonal tree for P1 by colony DNA sequencing (Figure 2d), demonstrate that our filtering approach is reliable and allows for the analysis of mitochondrial variability in the absence of a DNA-based reference.

Response Figure 4: Calling of mitochondrial somatic variants in the absence of a DNA-based reference. Corresponding to Supplementary figure 4. See above for legend.

4) In Figures 4 and 5 the authors provide evidence about how clonal tracking information can enhance single cell transcriptomics data. The authors state:

“Together, these results demonstrate that clonal tracking of single-cell transcriptomes can identify the molecular effects of clonal evolution in human, orthogonal to the use of mouse models^{34,37–39} but with consistent results.”

The consistency of results obtained from a limited group of heterogeneous patients is unclear. To truly make such a strong statement, the authors would need to examine many additional acute myeloid leukemia samples.

We agree with the reviewer and apologize for the overstatement. The point that we wanted to make is that the technological advances made by single-cell multi-omics and the specific approach presented in

the manuscript brings us a step further in the attempt to study molecular effects of clonal evolution in humans, which has previously been possible in mouse models only. We have now rephrased our statement to:

Lines 311-313: In sum, we have used a small heterogeneous patient cohort to demonstrate, as a proof-of-concept, that the acquisition of specific mutations is frequently linked to an altered gene expression program, which is consistent with data obtained from mouse models.

Lines 324-326: Taken together, these results demonstrate the ability of MutaSeq and mitoClone to delineate developmental and molecular defects of clonal evolution caused by leukemic and pre-leukemic mutations.

5) Fig. 2e/f displaying the results of the PhISCS approach appear to imply an order on how the various mutations were acquired. Many mitochondrial mutations however appear to be shared within a single clone and are otherwise not subclonal. It is not intuitive to follow how a temporal order may therefore be readily delineated here?

We apologize for this confusion. We did apply a statistical approach to group mutations into clones (now added to the methods, line 552-566 and available in the mitoClone package), based on which we color-coded the tree. Within a clone, we do not see strong statistical support for a specific order. We have therefore grouped the mutations from each clone together in figures 2e,f.

6) Some parts of the introduction and discussion make inaccurate claims that should be corrected. For instance, the authors state, "However, the latter studies have focused on analysis of single-cell ATAC-seq data..." However, the value of mitochondrial mutation calling in Smart-Seq2 has already been shown in Reference 31.

We thank the reviewer for pointing this out. As detailed above, we have now better clarified the advance over reference 31 (now ref. 33). This statement has been rephrased to:

Lines 86-89: However, the application of these methods to characterize leukemic stem cells has not been demonstrated, and in particular requires the ability to reliably detect clonal expansion events, associate clinically-relevant coding mutations to clones with high confidence, and draw statements on gene expression changes between clones.

As also summarized in reply to other reviewer comments, we have critically reviewed all claims drawn by the manuscript, and either added further evidence to support them, or chosen alternative formulations.

Reviewer #2 (Remarks to the Author):

In this work, Velten et al describe a modified version of the SmartSeq2 protocol, MutaSeq, that is able to define clones and associate targeted mutagenic information without affecting the overall performance of the original protocol. They analyse four AML patients using this methodology to define leukaemic, pre-leukaemic and healthy clones and use the information to get a better understanding of the biology of these clones.

We thank the reviewer for this summary of our work.

General Comments

The authors carefully looked into the implications of introducing additional oligonucleotides in the amplification reaction for the overall results of the SmartSeq2 protocol and followed an interesting approach by associating genomic mutations to mitochondrial mutations. However, there are certainly important limitations and concerns:

1) MutaSeq improves the dropout rate of SmartSeq2 but the dropout rate is still very high for a large number of regions (as depicted in figure 1e). The detection rate will strongly depend not only of the expression levels of the gene (figure 1d), but also of the distance of the mutation to the polyadenylation site and the complexity of the sequence of the gene of interest. The high dropout rate makes the association of the genomic and mitochondrial mutations difficult.

We agree that there is substantial dropout, and we tried to openly discuss and analyze this point throughout the manuscript (line 152-157):

While some statements on clonal hierarchies could be drawn solely based on calls of these nuclear somatic mutation (Figure S3a), the relatively high dropout of these sites impeded robust assignments of cells to clones (Figure S3b-d). Moreover, the result was biased by the expression levels of the mutated genes of interest: In cells with low expression, dropout was higher, leading to a higher fraction of false negative calls, i.e. false classifications of mutant cells as reference (Figure S3c-d and see also figure 1e).

Our data demonstrate that beyond expression level, the type of mutation (frameshift, nonsense or missense) affects dropout (line 123-124). Although our data do not provide statistical evidence for an effect of distance from poly-A site or sequence complexity on dropout, these are certainly valid points and we now mention them explicitly (line 124-126).

We agree that the association of the genomic and mitochondrial mutations can be challenging, especially if the target gene is not highly expressed or if the mutation causes degradation of the transcript. However, in many cases making these association is possible. With the exception of two frameshift mutations, all mutations shown in Figure 1e are significantly associated with mitochondrial mutations (see updated Figure S5a, also shown below). Further evidence is provided in the response to specific point 9 below. Finally, we take into account dropout rates in the statistical model used for inferring clonal structure (adapted from Malikic et al., Genome Research 2019, now explicitly referenced in the main text, line 177). Thereby, we are able to confidently associate individual cells to clones and perform quantitative analysis of clone-specific gene expression. We have now emphasized the point that statistically confident clonal assignments are only possible with MutaSeq when mitochondrial somatic variability is present in introduction, results and discussion:

Line 91-93: *Here, we introduce MutaSeq, a workflow that amplifies nuclear mutations from cDNA, and mitoClone, a computational tool that achieves high-confidence **clonal assignments and de novo discovery of clones using mitochondrial marker mutations when available.***

Line 196-198: *Taken together, our approach allows for the identification of putatively leukemic, pre-leukemic, and healthy clones and can assign cells to clones with high confidence, **if mitochondrial somatic variation is present.***

Line 347-348: ***In the presence of mitochondrial somatic variability,** our approach does not rely on previously known nuclear mutations to detect clones.*

Line 403-406: *In the absence of mitochondrial somatic variation, the use the MutaSeq protocol can be used to draw qualitative statements on clonal differentiation capacities (similar to ref. 28, and with an improvement over smart-seq2), but due to dropout neither enables the statistically confident assignment of cells to clones, nor differential expression testing between clones.*

Response Figure 5. Statistical association between nuclear and mitochondrial mutations. Included in the manuscript as Figure S5a. P-values are from a two-sided Fisher test.

2) The co-detection of genomic mutations is low (4-5 mutations out of 14 per cell according to figures 1c and S1a). The authors do not show if this is due to technical performance of the primer pool or due to coverage.

To answer this question, we targeted genomic sites on 13 highly expressed genes using our MutaSeq protocol and measured the co-detection of these sites in 48 cells. MutaSeq amplicons from all 13 genes were co-detected in over 97% of cells. We added supplemental figures which illustrate the enrichment of the target amplicons over background (Fig. S1f,g). Thus, we conclude that low expression levels are the

major driver of target-site dropout, but that the protocol as such does not introduce limitations regarding the ability to co-detect mutations.

Response Figure 6: Co-detection of amplicons from highly expressed genes. Included in the manuscript as figure S1f,g. **f.** Primer pairs were designed surrounding randomly selected sites on 13 highly-expressed genes in K562 cells (Table S5). The MutaSeq protocol was then performed using these primers on $n=48$ K562 cells. For each gene, the number of reads from MutaSeq amplicons (i.e. complete matches) is shown, after subtracting the average coverage of the surrounding areas outside of the targeted site (i.e. potential background signal). Seven cells with poor alignment rates (below 50%) were removed. **g.** Amplicon counts for 13 genes across 41 cells is shown as boxplots. The points in the overlaid beeswarm plot represent cells. Same underlying data as used in Figure S1f.

3) Previous knowledge is required for the detection of the specific genomic mutations to be analyzed. However, it is worth mentioning that MutaSeq would allow retrospective analysis of additional mutations.

We thank the reviewer for this suggestion and have now added a new subchapter demonstrating this ability (lines 200-224) and created a new figure S6.

4) The method may be difficult to be upscaled to droplet methods, which are strongly biased towards either 3' or 5' regions, relying in more expensive technologies such as SmartSeq2.

We agree with this point. This is a general problem of all existing single-cell RNA-seq clonal tracking protocols that enable a high confidence assignment of individual cells to clones (in particular, also TARGET-seq and other mitochondria-based approaches). We now spell this point out clearly in the discussion (line 406-410):

The second limitation of MutaSeq is its relatively low throughput. This limitation is currently shared with Smart-seq2 and alternative single-cell RNA-seq methods allowing high-confident assignment of cells to clones^{31,33}. Future work will focus on the inclusion of full-length coverage of the mitochondrial genome in droplet-based single-cell RNA-seq platforms.

5) It requires relative deep sequencing (~788,000 reads per cell).

We have further analyzed the sequencing requirements of MutaSeq, and found that while nuclear mutation calling requires deep sequencing, the basic identification of a clonal structure using mitochondrial reads does not. We have added a supplementary figure (S5c-e, see also below) and highlighted the findings in the main text (line 181-185):

Unlike genomic mutation calling from cDNA, identification of clonal identities from mitochondrial mutations (...) is possible at lower sequencing depths (Figure S5c-e), since mitochondrial genes are consistently highly expressed.

Deep sequencing is beneficial in the context of LSC characterization for other reasons as well, as we now discuss (line 370-374):

Deep transcriptome sequencing. In some cases, the bulk of leukemic cells displays gene expression signatures highly similar to stem cells, as observed here for the CD34+ blasts of patient P1. The

differences between LSCs and residual healthy HSCs are even more subtle. Previous work using shallow, microwell based sequencing of AML cells²⁷ has missed differences between LSCs, CD34+ blasts and residual healthy HSCs.

Response Figure 7, included in the manuscript as figure S5c-e. **c.** Effect of read depth on mitochondrial clusters. Clusters obtained from mitochondrial sites only were computed at full read depth (row “Original clusters”) and are compared to clusters obtained using the same methodology from data were single cells were down-sampled to read depths of 500k, 100k, or 20k per cell (“Downsampled”). Data from Patient 1 is shown. **d.** Like panel c, but for patient 2 (P2). **e.** Original clustering result and down-sampled clustering result are compared quantitatively using the Rand index.

6) The authors assume that cancer stem cells cannot be separated from healthy stem cells based solely on gene expression using a healthy donor as a reference. The authors should investigate if a focused analysis of the HSPC compartment would be able to separate the cancer stem cells from healthy stem cells in the different patients without relying on mutational information and then compare with the mutational information.

We thank the reviewer for this suggestion. We have now included a focused analysis of the HSPC compartment in the results section (line 261-262 and new figure S7f, also shown below).

Our analysis demonstrates that based on gene expression alone cancer stem cells cannot be reliably separated from healthy stem cells in an unsupervised analysis: the healthy and cancerous clones highlighted in the left panel of the figure below do not separate well. Other processes, such as the onset of differentiation and cell cycle, dominate gene expression variability in this compartment (right panels in the figure shown below). These processes may be altered upon transition to a cancer state, as in the case of patient 2 (see main figure 5a), but this is not always the case. Therefore, the use of mutation-derived clonal labels is required to draw meaningful statements about the distinct populations.

Response Figure 8, included in the manuscript as figure S7f: Unsupervised analysis of gene expression data from HSPCs does not distinguish between clones. **f.** uMAP representation of all Healthy HSPC-like cells from patients P1, P2 and P4. Data from these cells only were integrated using MNN62 and visualized in two dimensions using uMAP. Left panel: clonal identity is highlighted, see also Figure 4e. Right panels: Point color represents the expression of genes involved in differentiation (MPO for myeloid differentiation, FCER1A for erythroid/megakaryocytic differentiation) and cell cycle (TYMS, MCM2).

The authors state that the method does not rely on previously known nuclear mutations to detect clones. However, although the statement is technically true, they could not define clones using the mitochondrial mutations in 2 of the patients (50% of the analysed samples!!). Additionally, the association to the genomic mutations is not great because of the dropouts (see below).

We agree, and have made this limitation more explicit in the introduction (line 91-93), discussion (lines 347-348 and lines 402-403) and results (lines 196-198) sections. Briefly, interindividual variation is an important aspect of these analyses. However, we would also like to emphasize that even in the absence of mitochondrial variability, qualitative statements can be drawn, to a comparable extent as in GoT-seq

albeit at lower throughput (Nam et al., ref 28; see discussion, line 404). Regarding the association of genomic and mitochondrial mutations, see our reply to general comment 1 and specific comment 9.

Overall in the manuscript, the Figure legends are quite bare and not very descriptive which makes the interpretation difficult and more laborious. A much more detailed description of the figures is needed in the figure legends, clearly describing what is shown in each panel and how the data was obtained. Definition of the scales in the plots is also typically absent. The output for all the comparisons performed in this work should also be included (e.g. comparison of CD34- blasts to HSC/MPP-like cells), all data relevant to the article should be deposited (including DNA exome results for colony assays in P1) and the code must be made available.

We greatly appreciate the effort this reviewer has put into helping us make the manuscript better, more legible, and reproducible. We have implemented all suggested edits and when required provide further explanations below. We have also paid special attention to figure legends and definition of scales. All code underlying the analyses is now available in the *mitoClone* package, available from <https://github.com/veltenlab/mitoClone> which is sufficient to reproduce the major analysis and figures presented in the manuscript.

Specific Comments

1) The authors indicate in the introduction section that “previous protocols suffer from excessive dropouts, are of low throughput and /or require prior knowledge of genomic mutation sites”. This is also the case for this method as well.

The point we were trying to make here, is that our approach presents a solution to the problem of dropout since clones can be detected based on mitochondrial mutations. These are efficiently covered. The clonal identity of each cell is therefore specified at high levels of statistical confidence (Figure 2, S3b). Knowledge of genomic mutation sites associated with these clones is not required for the detection of clones and can in some cases be identified *de novo* (Figure S6 and see also general points 1 and 5); this information is only used to annotate clones as leukemic or pre-leukemic. To stress more clearly that we are not addressing the problem of dropout per gene, but rather the confidence of clonal assignments, we have reformulated this statement to

Line 83-84: *However, these protocols suffer from a lack of confidence in assigning cells to clones and/or require prior knowledge of genomic mutation sites.*

2) S1b should be plotted in a normalised way, to be able to compare how many reads are on target (for example in reads/100,000 of mapped reads) per cell.

This was the case. We clarified the figure legend and axis label in S1b.

3) Figure legends of figures S1c and S1d should explicitly indicate the meaning of the concentrations.

The change was implemented.

4) Figure 1e.- Define “reference” vs “mutant”. Why does the proportion of the call of the mutation change drastically according to the method used in some genes, such as SRSF2? How confident are the authors that the “reference” or “mutant” call is not affected by dropouts of the other allele?

Definitions have now been added to the figure legend. As we discuss in line 154-159 of the manuscript, reference calls are potentially dropouts of the mutant allele. This observation motivates the use of mitochondrial mutations for reconstructing clonal trees. Furthermore, the possibility of dropouts occurring is explicitly accounted for by the statistical method used for identifying clonal hierarchies and assigning cells to clones (adapted from Malikic et al., Genome Research 2019).

In the case of SRSF2, coverage with the MutaSeq protocol is very high, i.e. in most cells both the mutant and the reference allele are observed. By contrast, in the Smart-seq2 protocol, individual alleles frequently drop out. This explains the discrepancy between the proportions. Estimates of allele frequencies are similar in the two methods (figure 1f) since they are based on read counts.

5) Why is it that MutaSeq and SmartSeq2 are drastically over estimating some of the point mutations in Figure 1f?

The accuracy of the two methods in estimating allele frequency depends on the coverage of the site of interest. We have added this important co-variate to Figure 1f. There is no systematic bias for an over- or under-estimation of allele frequencies by either method, except for frameshift mutations, which are underestimated (see next point).

6) The authors state that both methods underestimate the abundance of frameshift mutations, likely due to nonsense mediated decay. Do the authors actually have evidence for this claim?

We do not provide experimental evidence for this statement. 'Likely' was replaced with 'possibly' and a reference to a literature report investigating this phenomenon in more detail (pubmed ID: 30032986) was included.

7) The authors should tone down the statement "MutaSeq efficiently covers both genomic target sites and the mitochondrial genome in single-cell RNA-sequencing experiments". The method improves the results when compared to SmartSeq2 but it has a high dropout rate in a large number of genes.

We agree and the statement has been changed to:

Together, these results demonstrate that MutaSeq efficiently covers the mitochondrial genome in single-cell RNA-sequencing experiments and provides improved coverage of genomic target sites compared to SmartSeq2.

8) The authors define preleukaemic and leukaemic mutations but they should better explain/demonstrate/reference the rationale behind the assignment for each mutation. Also, the first description of the status affiliation of the mutations appears at the very end of heading "Simultaneous mapping..." but there are references to this classification and separation much earlier in the text. Please clarify earlier on to gain in clarity of the article. The authors also mention that leukaemic cells often exist in a cell state that is similar to healthy HSC/MPPs. Could it be because these are cells that contain the mutation but they are not transformed into leukaemic cells?

We have added background information to the introduction:

Lines 75-77: Pre-leukemic stem cells are thought to typically carry mutations associated with CHIP (for example, in Dnmt3a) but not mutations associated with leukemia (for example, in Npm1)^{7,20,21}, potentially enabling their identification by profiling both known leukemic and known preleukemic mutations.

We then make clear that the labeling of clones as leukemic or pre-leukemic is based on previously known mutations when we first introduce the patients in the results part (lines 148-152):

*Bulk exome sequencing of the patients had identified **known** pre-leukemic mutations present at high allele frequency and **known** leukemic mutations present at a somewhat lower allele frequency (Figure 2a, Table S1: Patient 1: SRSF2, TET2/CEBPA and SRSF2, TET2/KLF7; Patient 2: DNMT3A/NPM1; Patient 3: SRSF2, IDH2; Patient 4: leukemic Trisomy 8 and BRAF mutations).*

Finally, we make clear that this assignment is in line with the functional definition of pre-leukemic (i.e. a clone preceding leukemia formation that is still capable of healthy blood production):

Lines 187-190: Across the patients, we identified clones carrying known pre-leukemic mutations (e.g. SRSF2, DNMT3A) and sub-clones carrying known leukemic mutations (e.g. CEBPA, NPM1) (Figure 2e,f). Below we definitely characterize these clones as leukemic or pre-leukemic based on their contribution to healthy blood production (see Figure 4).

We find the idea that cells may have acquired mutations but not yet transformed interesting, however it is beyond the scope of our study to investigate this point further. The view dominating in the field is that leukemic cells do exist in a stem-cell like state, whose differentiation is blocked at a later stage. Our data do not provide evidence contrary to that view.

9) The detection of the DNMT3A mutation present multitude of dropouts and the association values to the mitochondrial mutations are very low. Therefore it is difficult to make any statements or claims about the preleukaemic status of cells in patient P2.

We thank this reviewer for raising this point, which we would like to clarify. Despite the significant dropout in DNMT3A, we can draw statements for the following reasons:

Firstly, the clonal hierarchy (with a NPM1 wildtype parental clone and a NPM1 mutated subclone further marked by the mitochondrial 5999 T>C mutation) is clearly significant, according to the statistical criteria described in lines 552-566 of the methods section. Therefore, the parental NPM1 wildtype clone is preleukemic (a clone preceding leukemia formation).

Second, DNMT3A is one of the best characterized 'pre-leukemic' mutations associated with clonal hematopoiesis. While it may suffer from high dropout, its association with both the 'pre-leukemic' and the 'leukemic' clone, but not the 'non-leukemic' clone, is significant, see figure below. We have improved visibility of the mutant cells in figure 2c.

Response Figure 9: Number of reads of the mutant allele plotted against clonal identity. In the main text, clone 2 is labeled as 'pre-leukemic' and clone 3 is labeled as 'leukemic'. P values are from a wilcoxon test.

10) It is unclear in the main text which cells were analysed from which patient. It would be useful to have in the main text a brief description of the cells sorted and the quantity of each type used for the analysis.

We agree and have more extensively described the cell types included from each patient (line 142 – 145). A full overview is presented as figure S2.

11) Include percentages in the boxes in Figure S3a.

Percentages are now included (now Figure S2a).

12) Why weren't all populations in Figure S3a analysed for the 4 patients? E.g. Cd34-Lin+ cells were not analysed for P1 or P4? Why several of the gates are different between the healthy control and patients, such as FSC vs SSC (healthy) or CD34+ gate for patient 3. Can the authors explain the rationale?

The healthy control study (Velten et al., 2017) serves as a reference for CD34+ stem and progenitor cells. Since the Velten et al. 2017 study focused on HSPCs, SSC^{hi} cells, corresponding to mature granulocytic cells, were excluded. For this study, we aimed to include both immature and mature blasts with SSC^{hi} phenotype, and therefore expanded the SSC/FSC gate to include these.

Of further note, the healthy control sample was sorted on a different flow cytometer with slightly different settings, making fluorescence values difficult to quantitatively compare to the AML samples.

The lineage cocktail used here contained CD4, CD8, CD19, CD20, CD41a, and CD235a and therefore labeled mature T cells, B cells, erythroid, and megakaryocytic cells, but also leukemic cells aberrantly expressing these markers (e.g. some of the blasts express CD4 at variable levels). Hence, the lineage gate can be highly populated by T cells or blasts (see new figure S2d), depending on the CD4 expression state of the blasts, which differs between patients. We originally attempted to include Lineage positive cells from all patients, however, in P1 and P4, no such cells were covered, possibly due to a low abundance relative to leukemic cells. The only effect of the lineage gating scheme used for P1 and P3 on our data is that the representation of blast subpopulations may be skewed towards CD4-expressing blasts in these patients. Our manuscript does not draw quantitative statements on the composition of the blast populations and therefore the differences in sorting do not affect the outcome of this study. A brief discussion of this aspect was added to the main text (line 142 – 145) and methods (line 444-445).

Finally, the Lin-CD34+ population in P3 appears shifted to higher lineage expression values. Inconsistent expression patterns of surface markers are highly common in AML (see van Dongen et al., Leukemia 26, 1908-1975).

13) An additional UMAP plot should be added to the supplementary figures where cells are colored according to the gating strategy for each patient.

We thank the reviewer for this suggestion and have added the plot (Figure S2d).

14) Please indicate the methodology followed to obtain the clusters in figures 2b, 2c and 2d.

We have updated the methods section (lines 552-566) and provide the code for clustering mutations into clones as part of the mitoClone R package.

15) Could the authors show an example of how the data looked without the mitochondrial mutations included (Fig 2b and d)?

The data from figure 2b without the mitochondrial mutations included is shown in figure S3a. The data from figure 2d without the mitochondrial mutations included is shown below. In both cases, the same basic tree structure (of a parental clone carrying the SPEN, SRSF2 and EAPP mutations and two sub-clones carrying the CEBPA and KLF7 mutations, respectively) is identified by PhISCS, but the confidence of assigning individual cells is lower, due to dropout. This observation motivates the use of mitochondrial markers.

Response Figure 10: Heatmap of allele frequencies of nuclear mutations from the colony DNA-seq experiment (figure 2d). Clustering was obtained using the mitoClone default workflow.

16) The authors claim that there are mutually exclusive mutations such as KLF7 and CEBPA. However, it is difficult to define the mutation status of KLF7 in P1 and P3 (or TET2 and even SPEN in other patients) due to the high rate of dropouts (as expected from data in Figure 1). How certain can the authors be of this claim?

This appears to be a misunderstanding, mutations in KLF7, CEBPA, TET2 and SPEN are only observed in P1 based on exome sequencing (see table S1). Within this specific patient, the KLF7 and CEBPA mutations were exclusive to each other both in single-cell RNA sequencing (figure 2b, S3a) and in the sequencing of single cell derived colonies (figure 2d). We have re-written this paragraph and removed the reference to mutual exclusivity of these mutations to avoid ambiguity. Also note, that although these mutations are mutually exclusive in this particular patient, we make no claim that this observation is broadly applicable to AML in general.

17) The reference or demonstration of the fact that only bona fide mitochondrial mutations are specific to individuals should be shown.

We apologize for phrasing this statement as a known fact. What we meant to say was that we developed a filtering strategy on the assumption that bona fide mutations are specific to individuals, and we then demonstrate that this filtering strategy works in practice, using several validation experiments and control datasets. We have now made this clear (lines 163-166) and added further validations of the filtering strategy using a number of control datasets (lines 167-174 and new supplementary figure 4)

18) The description of the analysis of mitochondrial mutations is not completely clear. In particular, I find confusing the sentence "Individual cells were then called as mutant in these sites if at least 10% of the reads from that cell supported a minor allele.". Please clarify.

This point has been clarified in the methods section (lines 527-536).

Filtering of mitochondrial variants. To identify relevant somatic variants, we implemented the `mutationCallsFromCohort` function. In short, we select coordinates in the mitochondrial genome containing at least 5 reads each in at least 20 cells. To distinguish RNA editing events and true mitochondrial mutations, we then identify mitochondrial variants that occur in several individuals.

Individual cells are therefore called as “mutant” in a given genomic site if at least 10% of the reads from that cell were from a minor allele (i.e. distinct from the reference). Mutations present in at least 1% of cells in a given patient, but no more than 10 cells in any other individual, are then included into the final dataset and counts supporting the reference and mutant alleles are computed as for sites of interest in the nuclear genome. Mutations present in several individuals are stored as a blacklist and were used further for filtering some of the data analyzed in Figure S4.

19) The information relative to the genomic mutation status for TET2 and SPEN is not used in Figure 2 for P1. Was this due to the very high rate of dropouts?

The dropout rate of these mutations was very high and they were therefore not included to compute the clustering. However, we have now added them as labels to the heatmap in figure 2b since coverage is sufficient to demonstrate that they occur exclusively as mutant in the pre-leukemic and leukemic clones.

20) The “healthy” clone of patient P1 is clearly negative for preleukaemic mutation SRF2 but the cells for this clone also have the common characteristic of being mostly dropouts for Cebpa. It would be interesting to increase the number of SRF2-negative colonies in the DNA-seq (Figure 2d) to fully verify that they are CEBPA-negative.

This is a good suggestion. Unfortunately, no material from this patient is available to implement the suggested experiment (we typically receive 3-4 vials per patient from which we perform exome sequencing, single-cell RNA sequencing, and any follow up experiments such as colony sequencing). The cells in question are mostly CEBPA dropout because there are many T cells in this population; T cells don't express CEBPA, and don't give rise to colonies in these conditions, explaining the discrepancy in the size of this clone between colony and RNA-seq. The evidence for CEBPA being mutated downstream of SRSF2 is clear (any alternative hypothesis would require that the CEBPA mutation was acquired before the SRSF2 mutation, and later lost again; a highly unlikely scenario for which our data presents absolutely no evidence).

21) The legend of figure S2e is not complete.

Apologies. The relevant figure legend has been updated (now figure S4b)

22) Key legends in figure S2f are swapped.

We thank this reviewer for noticing this mistake. The key order is now corrected (now figure S4c). To clarify: In the left panel, sites were stratified by RNA mutation calling on the x axis and are color coded according to the result of DNA based mutation calling, while the right panel stratifies sites by the result of DNA based mutation calling on the x axis and color codes according to RNA based mutation calling.

23) Add the meaning of grey dots in key legend of figure S2g.

The label was added (Now figure S5a).

24) What cell type is the pale yellow cluster in Figure 3a?

These cells correspond to a sub-cluster of CD34+ blasts, which appear to constitute an intermediate state between Healthy-like HSC/MPPs and the population with the darker yellow labels. We have now labeled the two CD34+ blast clusters AP1 high and AP1 low, based on the expression of FOS/JUN genes. Since the high similarity of AP1 low CD34+ blasts and HSPCs impedes a clear separation of cancerous and healthy gene expression states here, the following analyses focus on the EGR1 high CD34+ blasts cluster, or the HSC/MPP cluster.

25) The population within Fig 3a labelled as “unclear” would be better labelled as “unknown” or “unclassified”.

The label is now changed.

26) It is difficult to distinguish HSC/MPP population from others in the healthy donor in Fig 3c. Please use a different colour.

The color is now changed.

27) It is unclear the cells that were used to perform the comparison in Figure 3g. Are the cells within the pale yellow cluster included in the category “CD34+ blasts” or only the cells within the dark yellow cluster? Please specify and justify the selection. In a similar way, can the authors specify which cells were used for the comparisons in figure 3h? According to Table S3, 162 genes obtained a FDR < 0.1; however, the Venn diagram in Figure 3h shows 639 genes. How is that possible? According to the Venn diagram in Figure 3h, there are much

fewer upregulated genes for Cd34- blasts. Is this population then more similar to the HSC/MPP? What about the downregulated genes?

Regarding the choice of population, see our reply to the reviewer's specific point 24. An explanation was added to the figure legend.

Figure 3g vs. Figure 3h (now figure 3i) analyze different aspects of the data. Figure 3g performs a focused comparison of only CD34+ EGR1^{high} blasts against HSC/MPP-like cells, in order to identify genes that are different between these highly similar populations. Figure 3i identifies 'highly expressed' genes for each population by comparison with all other populations in the dataset so as to identify genes that are shared by all blast populations compared to all other cell types present. We have now made this clearer in the text and the figure legend, and also corrected a labeling mistake (labels "62" and "581" were flipped; CD34+ blasts are more similar to the HSC/MPP population).

28) Where is the unsupervised analysis to separate HLF+, GATA2+, Flt3+ cells? Please show the output of the clustering. Also include RNA expression of genes HLF and Flt3 (in addition to surface marker) in fig S4a.

A summary of the unsupervised analysis is shown in figure 3f, using a visualization that allows stratification by patient. We have now included a display of the clustering highlighting RNA expression levels of the relevant genes in figure S7f.

29) Gene name S1008 should be included in text as gene for calprotectin.

Gene name was added.

30) It is very difficult to appreciate the differences in expression levels in Figure S4a. A different colour code should be used.

We thank the reviewer for pointing this out. The color code was changed (now figure S7a).

31) Would it not be better to show a specific marker which highlights the identification of each lineage in FigS4a? Along the same lines, Tal1 can be removed as very few cells seem to express this transcription factor.

The set of marker genes included in Fig S7a was changed to use markers more representative of single lineages, and this information was added to the panel labels. We further added callouts to Fig. 3e,h and S7a-c,f in the context of lineage identification since information contained in these figures also justifies the labels used in figure 3a.

32) The statement "Furthermore, all patients retain cells highly similar to healthy HSCs." should be modified to reflect that it is to a different degree in the different patients.

We thank the reviewer for pointing this out. The statement was changed (*Furthermore, all patients retain cells highly similar to healthy HSCs, although with variable abundance*).

33) Please confirm what the dashed line (eclipse) is highlighting in figures 4a to 4d. It is shifted in patients P3 and P4.

The eclipse serves as a guide to the eye to approximately highlight the location of the HSC/MPP population, which is accurately plotted on the uMAP in figure 3a. It was not used for any quantitative analysis. The figure legend has been clarified. The position in P3 and P4 was corrected.

34) The authors do not show sufficient data about their capacity to detect IDH2 nor NUP188, in terms of dropout proportion. The data in this respect in figures 4c and 4d is very difficult to appreciate. Please include similar data to figure 1e.

The data is included as new supplementary figure S8a (see also below) and was also added to the main text (lines 265-268):

In the absence of mitochondrial somatic variability, we used nuclear mutation calls in SRSF2, IDH2 and NUP188 for purely qualitative statements (Patient P3+P4). The capture rates of these marker sites ranged from 70% (SRSF2) to 11% (NUP188) (Figure S8a).

To further support the **qualitative** statements on P3 and P4 included in the manuscript, we here additionally underpinned them with quantitative arguments as follows. We refrain from including these analyses in the manuscript since they would unnecessarily disturb the read flow in a paragraph that focuses on qualitative aspects.

Lines 270-273: Clones associated with leukemic mutations were most prevalent in the blast compartments and were also detected in the HSPC compartment, but were almost absent in lymphoid (B, NK, T) lineages. In contrast, clones associated with pre-leukemic mutations were found in all lineages, but mostly displayed a decreased prevalence in lymphoid lineages (Figure 4e, S8b-d).

For P3: The p value from a fisher test for the hypothesis “among cells with a successful capture of IDH2, observations of the mutated allele are less frequent in lymphoid cells” is 2.3×10^{-6} . Covariates impacting the capture of IDH2 mutations (IDH2 expression, library quality as quantified by the number of genes observed) did not vary significantly between the groups.

For P4: Due to the low number of lymphoid cells seen in this patient, no significance is reached, but the observation contributes to the trend described ‘across patients’.

Lines 276-277: In patients P1 and P3, leukemic cells had retained the ability to contribute to the erythroid lineage. There is even an enrichment of IDH2 mutations in the erythroid precursors from P3 compared to all other cells from the patient ($p=0.01$ using a fisher test).

Response Figure 11, included in the manuscript as figure S8a. Bar chart depicting the percentages of cells covering the mutations used for annotating clones in P3 and P4.

35) In Figure 4c, it appears as if the higher allele frequency for the leukemic cells (as IDH2) is in the lymphoid cluster (T cells/NK cells) but figure 4e indicates that 50% of the cells in the T-cell cluster are non-mutated. How is this possible?

The color scale was flipped in figures 4c and 4d. We have corrected this mistake and apologize for the confusion.

36) The authors state that “Across patients, leukemic cells were not observed in lymphoid (B, NK, T) lineages” yet figure 4e clearly shows the presence of leukemic cells in the T cluster of cells patients P1, P2 and P3.

We apologize for the confusion. The paragraph has been rephrased (‘almost absent’, line 271)

37) How significant are the claims that CD96 is expressed higher in leukemic P3, when there are only approximately 8 cells within this category?

Despite the low number of cells, the statement “Independent of cell state, these cells further exhibited upregulation of CD96, which has previously been identified as a leukemia stem cell specific marker. CD96 was also highly expressed on leukemic HSC/MPP-like cells of patient P3, but not in patient P1, further illustrating the patient-specific nature of LSC markers (Figure S8e)”,

is supported by the following statistical tests:

- Wilcoxon test comparing Cd96 expression in (n=7) leukemic HSC-like cells from P3 vs. (n=20) leukemic HSC-like cells from P1: $p = 1.9 \times 10^{-6}$
- Wilcoxon test comparing expression in (n=7) leukemic HSC-like cells from P3 vs. (n=477) HSC—like cells from the reference individual: $p < 10^{-16}$ (healthy HSCs do not express this gene)

These comparisons were added to the boxplot in Figure S8e.

38) The authors need to take care, they describe the SRSF2 mutation as pre-leukemic but then subsequently state that “In patients P1 and P3 (SRSF2 mutated), leukemic cells”. I believe that the authors mean that P1 and P3 have the SRSF2 mutation, but it is misleading as currently stated.

The reference to genotypes was removed in order to avoid confusion.

39) The authors claim in P4 the ability to contribute to the erythroid lineage is restricted to pre-leukemic and non-leukemic clones, however figure 4e show the presence of leukaemic cells in the MEP/erythroid cluster.

There is a single NUP188 mutated CD34+ cell in P4 (see figure 4d), which falls into the MEP/erythroid cluster. We now do not comment on the presence of leukemic cells in the MEP cluster in P4:

Line 276-278: *For example, in patients P1 and P3, leukemic cells had retained the ability to contribute to the erythroid lineage, while in patient P2, this activity was restricted to the pre-leukemic and non-leukemic clones.*

And instead have briefly commented on the data from P4 in the discussion:

Line 343-346: *In patient P4, with one exception, only cells with a mature phenotype displayed leukemic mutations, illustrating that the disease can also be fueled by cells with a committed phenotype³⁸. Alternatively, the LSC population in this patient might be rare among CD34+ cells.*

40) The authors claim that in P4 mutations were only observed in CD34- blasts (and so is reflected in figure 4e) however, the UMAP in Fig 4d shows that the allele frequency for NUP188 is 1.0, with high log₁₀ read counts in the HSC/MPP region of the UMAP. Can the authors please explain this and clarify in the text.

The color scale was flipped in figures 4c and 4d. We have corrected this mistake and apologize for the confusion.

41) In Figure 4e it is important to add information about the total number of cells in each category for each patient.

A supplementary figure has been added (Figure S8b) that contains this information, and source data have been added. In the main figure 4e, cell types are only included if represented by at least 10 cells to avoid making misleading impressions.

42) Also in Figure 4e, I wonder if the category “Blasts (CD34- HBZ+)” is labelled correctly since according to figure S4a the expression of HBZ is very restricted outside some CD34+ blasts.

The intended meaning of this legend was “CD34- and CD34+HBZ+ blasts”. We have corrected this label.

43) Figure 4e shows that all cells in HSC/MPP compartment of patient P4 are “non AML”. However, figure 4d shows allele frequency close to 1 for NUP188 for most cells in the HSC/MPP compartment. How can this be possible?

The color scale was flipped in figures 4c and 4d. We have corrected this mistake and apologize for the confusion.

44) The authors should tone down the statement “Together, these results demonstrate that leukaemic mutations may initiate differentiation blocks at various levels.” Although the results suggest that this could be the case, no additional experiments to demonstrate such blockage in differentiation were performed.

We modified the statement to *Together, these results indicate that leukemic mutations may initiate differentiation blocks at various levels, as previously reported^{4,25,35}.*

45) In figure 5a, the text indicates “high” expression of HFL but data shows barely below 1. Please define the scale of expression.

The scale of expression (log-normalized according to Seurat package defaults) is now clarified in this and similar figures. Wording in the main text was changed to ‘relatively high’ expression since this statement derives from the comparison with the other groups. P-values for comparison were added to the figure.

46) In Figure 5a there seems to be a typo in the legend, since it seems to relate to patient P2 and not to patient P3.

The typo was corrected.

47) In Figure 5a, indicate number of cells for each category. It looks like there are very few cells which could affect the interpretation.

The number of cells was indicated. The interpretation is not affected, since all statements drawn from this analysis are supported by statistical tests. The p-values were added to the figure.

48) How many cells are involved in the comparisons showed in figure 5b? Are enough cells to reach enough

statistical power to make those comparisons? This is a general concern across all volcano plots shown in the article and should be addressed.

The number of cells per category was 55 pre-leukemic CD34+ cells and 50 leukemic CD34+ cells. In figure 5e, the number of cells per category is 105 pre-leukemic or leukemic and 41 non-leukemic cells. In figure 3g, 667 Healthy HSPCs are compared to 569 AP1-high CD34+ blasts. The number of cells included in the differential expression testing are therefore well aligned with the requirements of the test used (Finak et al., 2015). All numbers have been added to table S3.

49) It is difficult to appreciate which genes are differentially expressed in figure 5b. Because of the scale, it looks like Hoxb3/CD96 have a logFC close to 0. Indicate the thresholds in the figure and include full results as supplementary data.

For all volcano plots, all results are now included in supplementary table 3. Log fold change is only included in the figure to demonstrate if the gene is up- or downregulated and a guide was added.

50) The authors discuss the role of KLF7 mutation in HSPC cells from patient P1 (Fig 5d). How many cells are included in the each category of the comparison? Have they also looked at the CEBPa clone, as these cells also occupy the same cellular population/state?

The raw data underlying figure 5d and S8c,d is shown below and is now included as figure source data with the manuscript. Statistical significance was determined for all statements drawn, see asterisks and error bars included in the figure. The analysis of the CEBPA clone is shown in figure S8c and the results are briefly discussed in the main text (lines 271-273).

Cell type	Total number of cells	Of which pre-leukemic or leukemic	Of which leukemic (KLF7)	Of which leukemic (CEBPA)
B cells/precursors	18	3	0	0
CD34- Cells	116	77	25	14
CD34+ Blasts	880	851	373	236
HSC/MPPs	34	33	5	15
MEP/Erythroid	25	22	14	5
Mitotic HSPCs (G2/M)	13	13	10	2
Neutrophil precursors	6	4	0	3
NK cells	94	58	6	0
T cells	244	60	1	4

51) Include full results for the comparison of pre- and leukaemic cells. Indicate the number of cells in each category. Also, was it done by patient and/or merging patients?

This analysis was performed for each patient individually and only gave significant hits in the case of P2. These are now included in table S3. The number of cells per group is now indicated in the figure legend (105 pre-leukemic or leukemic and 41 non-leukemic cells)

52) It would be interesting to know how many cells were included in each category of the comparison of (pre-)leukemic vs non-AML cells. It is peculiar that so few genes came up as differentially expressed for P2 taking into account that these cells appear in very different locations of the UMAP. Also, it is curious that only FOS is upregulated out of the rest of AP1 factors.

This is an interesting point. The number of cells has been added to the figure legend. The effects that the mutations have on the position in the uMAP are analyzed elsewhere (i.e. Figure 4, 5a,d and S8c,d). The tests in figure 5b/e were performed while accounting for the cell type covariate, so as to identify changes in genes expression independent of transformations of cell state, as illustrated in figures 5c and 5f. Figure legends were clarified and the main text was amended to note that these results are 'independent of cell type'.

53) Please include in Table 1 the data regarding to which regions were targeted for patients P2 and P4. Describe the nature of highlighted genes in this table.

The information was included and the highlighting was removed.

54) Consistency is required throughout, the authors use pre-leukemic and preleukemic, one should be chosen and used throughout the manuscript.

We now use the term pre-leukemic consistently.

55) After multiple occurrences of a colon in the text a capital letter is used, this should be corrected.

We have corrected these instances.

56) The authors should consider to re-order the panels within the figures and in some cases the figures themselves to ensure that they are referred to in a chronological order throughout the text (for example figure S2c is mentioned after S2g, fig S3 is mentioned before S2).

Supplementary figures were now reordered to reflect their appearance in the text to the extent possible (some supplementary figures are needed in different contexts)

57) Maybe “Unanimously” should be swapped to “unequivocally”.

The term was replaced with ‘statistically confident’ or ‘confident’.

REVIEWER COMMENTS

Reviewer #1 (Remarks to the Author):

The authors have done an excellent job of addressing all of the concerns I had raised. The findings discussed in the paper are now much more clearly presented. The findings discussed in Figure 4 of HSC-like states among leukemic cells is very important.

Reviewer #2 (Remarks to the Author):

The revised manuscript has much improved. The addition of the mitoClone R package is indeed interesting and provides a useful tool in itself. In line with this, the additional validation for the unsupervised identification of clones based solely on the mitochondrial data by using mitoClone is certainly attractive since it allows the reanalysis of previously obtained datasets using SmartSeq2 technology, even if a link with the nuclear somatic mutation cannot be established.

Most of my comments have been addressed by the authors in the revised manuscript. However, there are still a few comments that remain unanswered:

1. Although the identification of clones is certainly useful, it is the association of the defined clones and (pre-)leukaemic mutations that results the most attractive. A clear limitation of the described method is the ability to detect some of the nuclear somatic mutations. The limitation may well arise from multiple sources such as low expression levels of the gene of interest, the stability of this RNA, the nature of the mutation (frameshift, nonsense or missense), the distance to the polyadenylated region or the intrinsic complexity/composition of the sequence. This is definitely a very important point that must be discussed in the section "Limitations" within the Discussion.
2. The figure legend of Figure S7f should be clarified to indicate that healthy and (pre-) leukaemic cells are included in the UMAP.
3. Figure S7f and Response Figure 8 are similar but not identical. Explain the discrepancies. This UMAP seems to be strongly influenced by cell cycle, as evidenced by the clear separation of cells by the expression of TYMS and MCM2. Although there are not many cells labelled as "Cancer", they visually look like they cluster together at the top left and bottom centre of the S7f plot (especially cells from patients P2 and P4). The authors should regress/minimise the effect of the cell cycle before obtaining the UMAP as it seems that it strongly influences the result.
4. Regarding the response to specific point 9). The authors enrol in a semantic argument. It is widely accepted that the term "preleukaemic" will apply to clones that present a mutation that predisposes to the acquisition of a leukaemic phenotype. In the case of P2, there is no dispute about the separation of NPM1 WT and NPM1 mutant clone (clones 2 and 3 in Response Figure 9). However, I am still not fully convinced of the association of these clones to DNMT3A. The plot in Response Figure 9 should also consider the number of WT reads associated to each clone to ensure that the increase of mutated reads in clones 2 and 3 is not associated to the actual number of cells in which the mutation is detected. Finally, this figure should be included in the Supplementary Figures in the manuscript.
5. The authors state in their rebuttal letter: "Our manuscript does not draw quantitative statements on the composition of the blast populations and therefore the differences in sorting do not affect the outcome of this study." However, lines 270-273, figures 4e and S8b-e clearly do that.

Reviewer #3 (Remarks to the Author):

In their paper "Identification of leukemic and pre-leukemic stem cells by clonal tracking from single-cell transcriptomics" the authors analyse leukemic stem cells and progenitors using a combination of mitochondrial and nuclear SNVs.

This is a paper of clearly great interest to the field. I was asked to

specifically comment on the reconstruction of clonal hierarchies and analysis of mitochondrial mutations.

Firstly, I thank the authors for describing the results of their evolutionary analyses as clonal hierarchies, which is what they are, not phylogenetic trees. In general, their approach is sound and considering that even by-eye informatics reveals the clonal hierarchies from the absence/presence matrix of SNVs quite clearly, one can be confident about the results of their method.

I have some comments and questions that might help to make the manuscript easier to understand for readers.

General comments:

The separation of RNA editing events from actual mutations seems reasonable. However, the authors state a 97% correct classification rate (l168). That should mean that approx. 3% of mutation calls are still RNA editing events. In Figure 2b P1 has one additional grey clone with only one mutation in a very few number of cells which were not found in colony seq. The numbers seem low enough for this to be caused by RNA editing events that escaped the filter. Can the authors comment on this please?

I read that "individual cells are therefore called as mutant in a given genomic site, if at least 10% of the reads from that cell were from a minor allele" (l531). That would suggest that the per-cell VAF now has a lower bound of 10%. However, in l541 the authors call a cell "mutant" based on a VAF cut-off of 5%. Please clarify this contradiction or explain the processing more clearly to the reader to avoid misunderstandings.

The clustering of mutations into clones and assignment of cells is a valuable addition. I'm wondering about the likelihood cut-off of "smaller than 1 per cell" (l566). This seems arbitrary and the documentation of the function in the R package does not provide further information. Shouldn't it be possible to create a situation where the merged clone and separate clones are two nested models and employ a likelihood-ratio test for rejection of the null "same clone"? The authors should at least discuss this cut-off or provide a rationale for it.

Minor comments:

l531 "in a given genomic site": I assume the authors mean site of the mitochondrial genome?

l562 (equation): sometimes it's $N_{\{cg\}}$, sometimes $N_{\{c,g\}}$ (same for $M_{\{cg\}}$). Please consistently use the comma (or don't).

Figure 2: The mutation mt:4693 T>C in panel f) is called mt:4639T>C in panel c). I assume a typo in panel f).

Throughout the ms: the authors introduce the leukemic stem cell (LSC) abbreviation in the intro but often (e.g. l87) the full word is used. I would use LSC throughout.

Reviewer 1

The authors have done an excellent job of addressing all of the concerns I had raised. The findings discussed in the paper are now much more clearly presented. The findings discussed in Figure 4 of HSC-like states among leukemic cells is very important.

| We thank the reviewer for their kind words.

Reviewer 2

The revised manuscript has much improved. The addition of the mitoClone R package is indeed interesting and provides a useful tool in itself. In line with this, the additional validation for the unsupervised identification of clones based solely on the mitochondrial data by using mitoClone is certainly attractive since it allows the reanalysis of previously obtained datasets using SmartSeq2 technology, even if a link with the nuclear somatic mutation cannot be established.

| We thank the reviewer for their kind words and detailed review.

Most of my comments have been addressed by the authors in the revised manuscript. However, there are still a few comments that remain unanswered:

1. Although the identification of clones is certainly useful, it is the association of the defined clones and (pre-)leukaemic mutations that results the most attractive. A clear limitation of the described method is the ability to detect some of the nuclear somatic mutations. The limitation may well arise from multiple sources such as low expression levels of the gene of interest, the stability of this RNA, the nature of the mutation (frameshift, nonsense or missense), the distance to the polyadenylated region or the intrinsic complexity/composition of the sequence. This is definitely a very important point that must be discussed in the section "Limitations" within the Discussion.

| We have now added this point to the limitations section as follows:

In the presence of mitochondrial somatic variation, an association between clones and nuclear mutations is only possible for mutations in highly expressed genes, and can be limited by the nature of the mutation (e.g. frameshift mutations) and possibly other factors such as sequence complexity.

2. The figure legend of Figure S7f should be clarified to indicate that healthy and (pre-) leukaemic cells are included in the UMAP.

| The first sentence of this figure legend has been changed to:

uMAP representation of all cells with a healthy HSPC-like gene expression signature from patients P1, P2 and P4. These include both healthy and pre-(leukaemic) clones, see figure 4.

3. Figure S7f and Response Figure 8 are similar but not identical. Explain the discrepancies. This UMAP seems to be strongly influenced by cell cycle, as evidenced by the clear separation of cells by the expression of TYMS and MCM2. Although there are not many cells labelled as "Cancer", they visually look like they cluster together at the top left and bottom centre of the S7f plot (especially cells from patients P2 and P4). The authors should regress/minimise the effect of the cell cycle before obtaining the UMAP as it seems that it strongly influences the result.

| We had unintentionally not copied the final figure S7f to the reviewer response. We had changed the layout and included a few more genes in the final figure. When looking into this analysis again, we found that the MNN algorithm (Haghverdi et al., 2018) used for this analysis makes use of random numbers, a behavior that we had missed before and that is also not documented in the fastMNN package. Hence, the arrangement of points on the final uMAP ended up slightly different (rotated, etc.) between different runs of the script. We now set an arbitrary seed (0xb33f) for the random number generator to ensure future reproducibility,

and have updated figure S7f accordingly. The other algorithms used in this manuscript either do not make use of random numbers, or a seed was specified (e.g. in the uMAP function). While there is a slight enrichment of cancer cells in some parts of the plot, this just indicates that leukemic mutations do affect the transcriptome, as discussed in the context of figures 4 and 5. It is clear that positions in the uMAP cannot be used to separate or identify cancer and healthy cells in the absence of clonal labels.

Cell cycle strongly changes as HSCs begin to leave the quiescent state and differentiate (see e.g. Passegué et al., J Exp Med 2005). Hence, strong changes of cell cycle in this compartment are not separable from differentiation processes, as described by several single cell studies previously (Tusi et al., Nature 2018, Giladi et al., Nature Cell Biology 2018, Velten et al., Nature Cell Biology 2017). It is therefore common practice in the field not to regress out cell cycle from the analysis. Due to the strong correlation of cell cycle genes and priming/differentiation genes, simply excluding known cell-cycle related genes (from the Seurat package, original reference: Tirosh et al., Science 2016) from the analysis only mildly affects the uMAP projection, as shown below.

Response Figure 1: uMAP representation of all cells with an healthy HSPC-like expression signature from patients P1, P2 and P4. These include both healthy and pre-(leukaemic) clones, see figure 4. Cell cycle associated genes (Tirosh et al., 2016) were removed from the dataset, data were integrated using MNN and visualized in two dimensions using uMAP. Left panel: clonal identity is highlighted, using the same strategy as in Figure 4e.

4. Regarding the response to specific point 9). The authors enrol in a semantic argument. It is widely accepted that the term “preleukaemic” will apply to clones that present a mutation that predisposes to the acquisition of a leukaemic phenotype. In the case of P2, there is no dispute about the separation of NPM1 WT and NPM1 mutant clone (clones 2 and 3 in Response Figure 9). However, I am still not fully convinced of the association of these clones to DNMT3A. The plot in Response Figure 9 should also consider the number of WT reads associated to each clone to ensure that the increase of mutated reads in clones 2 and 3 is not associated to the actual number of cells in which the mutation is detected. Finally, this figure should be included in the Supplementary Figures in the manuscript.

While the site of interest on DNMT3A (irrespective of mutant or wild type) is detected with at least one read in 24.5% of cells from clone 1, this is only the case in 11% of cells from clone 2 and 3. Still, mutations are only observed in clone 2 and 3. Hence, the observation of mutant reads in clone 2+3 is not a consequence of a higher read depth in these clones. We have modified the figure and statistical analysis to include covariate of total read coverage and now include it as supplementary figure 5b, also shown below. We have further expanded the figure to also include similar analyses for the SPEN T2324A and TET2 R1452X mutations and their association with the (pre-)leukemic clones in patient P1.

5. The authors state in their rebuttal letter: “Our manuscript does not draw quantitative statements on the composition of the blast populations and therefore the differences in sorting do not affect the outcome of this study.” However, lines 270-273, figures 4e and S8b-e clearly do that.

This statement discusses the implication of the use of slightly different sorting gates used in the different patients (Figure S2). A better wording is:
*Our manuscript does not draw quantitative statements on the **differential abundance of the blast populations between patients** and therefore the differences in sorting do not affect the outcome of this study. A brief discussion of this aspect was added to the main text (line 142 – 145) and methods (line 451-452).*

Figures 4e, S8b-e investigate the clonal composition of blast populations within single patients and are therefore not affected by the use of slightly different sorting gates in the different patients.

Reviewer #3 (Remarks to the Author):

In their paper "Identification of leukemic and pre-leukemic stem cells by clonal tracking from single-cell transcriptomics" the authors analyse leukemic stem cells and progenitors using a combination of mitochondrial and nuclear SNVs.

This is a paper of clearly great interest to the field. I was asked to specifically comment on the reconstruction of clonal hierarchies and analysis of mitochondrial mutations.

Firstly, I thank the authors for describing the results of their evolutionary analyses as clonal hierarchies, which is what they are, not phylogenetic trees. In general, their approach is sound and considering that even by-eye informatics reveals the clonal hierarchies from the absence/presence matrix of SNVs quite clearly, one can be confident about the results of their method.

| We thank the reviewer for their summary of our work.

I have some comments and questions that might help to make the manuscript easier to understand for readers.

General comments:

The separation of RNA editing events from actual mutations seems reasonable. However, the authors state a 97% correct classification rate (1168). That should mean that approx. 3% of mutation calls are still RNA editing events. In Figure 2b P1 has one additional grey clone with only one mutation in a very few number of cells which were not found in colony seq. The numbers seem low enough for this to be caused by RNA editing events that escaped the filter. Can the authors comment on this please?

Indeed, this is the only case where a clonal identity is determined based on a single site, with no further evidence from other mitochondrial markers or genomic mutations. We now explicitly discuss this clone in lines 228-232:

We also take note of a putative non-leukemic clone in P1 marked by a single mitochondrial variant (5492T>C). With one exception, all cells carrying this variant are positive for the T cell marker CD3 (Figure 2b, see also Figure 4a), but they display diverse TCR alpha and beta chain sequences (not shown). Hence, this variant was likely acquired in a T cell precursor, although we cannot formally exclude that it corresponds to a T cell-specific RNA editing event.

We take note that in animals, no tissue- or cell-type specific mitochondrial RNA editing has been described in literature.

For further clarification we now also highlight that the 97% classification rate is at the level of genomic sites (and not at the level of single-cell mutation calls), see line 167 and also the next point.

I read that "individual cells are therefore called as mutant in a given genomic site, if at least 10% of the reads from that cell were from a minor allele" (l531). That would suggest that the per-cell VAF now has a lower bound of 10%. However, in l541 the authors call a cell "mutant" based on a VAF cut-off of 5%. Please clarify this contradiction or explain the processing more clearly to the reader to avoid misunderstandings.

The two steps '*Filtering of mitochondrial variants*' and '*Construction of clonal hierarchies*' described in the methods have different aims.

The first step attempts to remove potential RNA editing events at the level of genomic sites. In that context, we observed that variants with a very low VAF were often not following clonal hierarchies, and likely correspond to RNA editing. Only for the purpose of removing these sites, we set a threshold that was relatively stringent (requiring at least 10% of variant reads). The output of that step is then a list of 'trustworthy' genomic sites.

The second step attempts to assign a genotype label of 'mutant' or 'non-mutant' to single cells at each genomic site that we trust to be real genetic variants and not RNA editing events. In this case, we assume that we are only looking at true genetic variants and not sites that are RNA edited. The only sources of noise therefore are sequencing errors or inter-well contamination, which occurs at very low rates (Figure S10). Hence, here we opted for a lower threshold (requiring only 5% of variant reads for a mutation call).

We have updated the methods (lines 534-554) to make this difference clear. Furthermore, in the package vignette '*Variant calling and blacklist creation*', we now provide additional practical recommendations on the filtering of mitochondrial variants and the choice of the parameters involved.

The clustering of mutations into clones and assignment of cells is a valuable addition. I'm wondering about the likelihood cut-off of "smaller than 1 per cell" (l566). This seems arbitrary and the documentation of the function in the R package does not provide further information. Shouldn't it be possible to create a situation where the merged clone and separate clones are two nested models and employ a likelihood-ratio test for rejection of the null "same clone"? The authors should at least discuss this cut-off or provide a rationale for it.

We agree that in theory, a likelihood-based test would be attractive for clustering mutations into clones. In practice, it is often beneficial to group mutations into clones even if there is some statistical evidence that a few cells display only one, but not the other mutation. For example, using reasonable assumptions on false positive and false negative rate, it is likely that there are a few cells in patient P1 which carry the CEBPA, but not the mt:2537G>A mutation (see figure 2a). However, for purposes of differential expression analysis between clones, and for providing interpretable, meaningful analyses throughout the manuscript, there is a clear benefit in merging them into one CEBPA mutated clone. The likelihood threshold of 1 per cell simply serves as a starting point to guide a biologically meaningful analysis.

The development of a more formal test would open unresolved statistical questions. When two clones are merged, the optimal assignment of cells to clones changes, depending on the choice of the false positive and false negative rate (see illustration below). The model employed in our manuscript (Malikic et al., Genome Research 2019) infers a matrix of ‘true’ mutational status per cell. Therefore, if two mutations are merged into a clone, the new model will have one free parameter less per cell. However, at the same time this matrix is subject to constraints so as to enforce a hierarchical relationship between the mutations. Hence, the difference in degrees of freedom between the models is non-trivial to estimate. Bayesian approaches such as described by Zafar et al., Genome Research 2019 and Jahn et al., Genome Biology 2016 might constitute a starting point for model comparison. In practice, however, the optimization of a likelihood through integer linear programming (Malikic et al., Genome Research 2019), which form the basics of the clonal hierarchy reconstruction employed here, has clear advantages: It always converges to the optimal solution, unlike MCMC approaches; it is faster; and it is easier to use.

We have adapted the manuscript (lines 565-586 and legend to figure 2e) to make clear that an arbitrary cutoff is used, and we now in the package vignette ‘Computation of clonal hierarchies and clustering of mutations’ provide further practical recommendations on the clustering of mutations.

Response Figure 3: Merging of clones results in different maximum likelihood assignments of cells to clones. $P(y|x)$ is the probability of observing y when the true status is x (e.g. observing a mutation when in reality only reference alleles are present). Due to constraints applied on the matrix x , an estimation of degree of freedoms between model 1 and model 2 is non-trivial.

Minor comments:

1531 "in a given genomic site": I assume the authors mean site of the mitochondrial genome?

| For clarity, the wording was replaced by ‘in a given site of the mitochondrial genome’

1562 (equation): sometimes it's $N_{\{cg\}}$, sometimes $N_{\{c,g\}}$ (same for $M_{\{cg\}}$). Please consistently use the comma (or don't).

| We now consistently don't use the comma.

Figure 2: The mutation mt:4693 T>C in panel f) is called mt:4639T>C in panel c). I assume a typo in panel f).

| The typo was fixed in panel f.

Throughout the ms: the authors introduce the leukemic stem cell (LSC) abbreviation in the intro but often (e.g. 187) the full word is used. I would use LSC throughout.

| We agree that this will improve consistency and now use LSC throughout.

REVIEWERS' COMMENTS

Reviewer #2 (Remarks to the Author):

The authors have satisfactorily addressed my comments.

I only have a minor remark related to my comment #3, regarding the regression of the cell cycle effect. The authors state that they removed genes associated to the cell cycle from the analysis. I believe that it would have been a better approach to obtain a cell cycle score using the cell cycle genes and regress out this score. However, I agree with the results presented by the authors.

Reviewer #3 (Remarks to the Author):

The authors have addressed all my concerns sufficiently. The findings in the manuscript are now more clearly presented. I thank the authors for their efforts.

Response to final reviewer comments

Reviewer #2 (Remarks to the Author):

The authors have satisfactorily addressed my comments.

I only have a minor remark related to my comment #3, regarding the regression of the cell cycle effect. The authors state that they removed genes associated to the cell cycle from the analysis. I believe that it would have been a better approach to obtain a cell cycle score using the cell cycle genes and regress out this score. However, I agree with the results presented by the authors.

We thank the reviewer for their detailed review. We consider removal of cell cycle genes, and not regression of cell cycle, to be standard practice in single cell analyses of bone marrow since differentiation and cell cycle are closely linked in this system (see Tusi et al., Nature 2018, Giladi et al., Nature Cell Biology 2018, Velten et al., Nature Cell Biology 2017).

Reviewer #3 (Remarks to the Author):

The authors have addressed all my concerns sufficiently. The findings in the manuscript are now more clearly presented. I thank the authors for their efforts.

We thank the reviewer for their review and helpful suggestions.